# Bridging scales in disordered porous media by mapping molecular dynamics onto intermittent Brownian motion

Colin Bousige [1], Pierre Levitz [2✉] & Benoit Coasne [3✉]

Owing to their complex morphology and surface, disordered nanoporous media possess a rich diffusion landscape leading to specific transport phenomena. The unique diffusion mechanisms in such solids stem from restricted pore relocation and ill-defined surface boundaries. While diffusion fundamentals in simple geometries are well-established, fluids in complex materials challenge existing frameworks. Here, we invoke the intermittent surface/pore diffusion formalism to map molecular dynamics onto random walk in disordered media. Our hierarchical strategy allows bridging microscopic/mesoscopic dynamics with parameters obtained from simple laws. The residence and relocation times – $t_A$, $t_B$ – are shown to derive from pore size $d$ and temperature-rescaled surface interaction $\varepsilon/k_BT$. $t_A$ obeys a transition state theory with a barrier ~$\varepsilon/k_BT$ and a prefactor ~$10^{-12}$ s corrected for pore diameter $d$. $t_B$ scales with $d$ which is rationalized through a cutoff in the relocation first passage distribution. This approach provides a formalism to predict any fluid diffusion in complex media using parameters available to simple experiments.

[1] Univ. Lyon, Université Claude Bernard Lyon 1, CNRS UMR 5615, Laboratoire des Multimatériaux et Interfaces, F-69622 Villeurbanne, France. [2] Sorbonne Université, CNRS UMR 8234, PHENIX Lab, 75252 Paris, France. [3] Univ. Grenoble Alpes, CNRS, LIPhy, 38000 Grenoble, France. ✉email: pierre.levitz@sorbonne-universite.fr; benoit.coasne@univ-grenoble-alpes.fr

Fluid diffusion in porous media involves complex phenomena arising from the restricted diffusivity imposed by the host porous geometry and the fluid/solid interaction[1–4]. While the medium morphology and topology impact the fluid dynamics at almost any pore lengthscale $d$, the effect of fluid/solid forces roughly scales with the porous surface to volume ratio $S/V \sim d^{-1}$ [5,6]. This leads to rich dynamics in nanoporous media (for which $d$ is of the order of the intermolecular force range $\zeta$) with intriguing aspects such as anomalous single-file diffusion, intermittent Brownian dynamics, stop-and-go diffusion with an underlying surface residence time, etc.[7]

For simple pore morphologies (e.g., planar, cylindrical), a unifying picture has emerged with well-identified dependence on temperature $T$, fluid density $\rho$, mean free path $\lambda$, pore size $d$, fluid molecule size $\sigma$, etc.[5,8,9] Single-file diffusion is restricted to $d \sim \sigma$ while the diffusion mechanism for $d \gtrsim \sigma$ depends on $\rho$ and $\lambda$; Knudsen diffusion for fluids with $\lambda \gg d$ and molecular diffusion for $\lambda \lesssim d$. For materials with large $S/V$, diffusion involves intermittent dynamics with subsequent surface adsorption and in-pore relocation steps[10,11]. When relocation is negligible (i.e. at low $T$ and/or $\rho$ where the pore center is depleted in fluid), diffusion is governed by surface diffusion described using the Reed–Ehrlich model[12,13]. In contrast, when relocation contributes to the overall dynamics (non-negligible pore center density), the intermittent Brownian motion is a rigorous formalism to upscale the local microscopic dynamics to any upper scale[14]. Diffusion in disordered porous media is far more complex as coupled geometrical and surface interaction effects lead to novel phenomena[15–17]. The fluid diffusion in such heterogeneous solids involves a non-trivial diffusivity landscape as surface diffusion/in-pore relocation boundaries are ill-defined. Diffusion in such rough landscapes is even more puzzling for nanoporous media as (1) the underlying propagators – i.e., the probability that a molecule moves by a quantity $r$ in a time $t$ – are not necessarily Fickian[18], and (2) non-vanishing surface interactions in the pore leads to self-diffusivity $D_s$ different from the bulk even far from the surface[19].

Due to the continuum hypothesis breakdown at the nanoscale[1,16], statistical mechanics is the appropriate formalism for complex diffusion in disordered media[7]. In particular, generalization of molecular intermittence to heterogeneous media using the Fokker-Planck or path integral formalisms allows linking microscopic to macroscopic dynamics[20]. However, while these approaches rely on available material parameters (e.g. porosity $\phi$, $S/V$ ratio, structure factor $S(q)$), fluid dynamics concepts such as surface residence, in-pore relocation, and their time constants are often used as guessed inputs (typically, relocation/surface diffusion are assumed to be Fickian with diffusivities equal or orders of magnitude slower than the bulk[14]). While this qualitatively captures the complex dynamics at play, there is a strong need to establish physical laws from simple parameters such as pore size $d$ and fluid/solid interaction strength $\varepsilon$. In this context, hierarchical simulations[13,21,22] allow upscaling the microscopic dynamics assessed from atom-scale simulations into kinetic Monte Carlo simulations; a precalculated free energy map $\Delta F(\mathbf{r})$ is used in a random walk approach with corrected hoping rates $k \sim \exp[-\Delta F(\mathbf{r})/k_BT]$[23,24]. However, extension to disordered solids is almost intractable because of their large representative elementary volume. Moreover, despite their robustness, such extensive simulations do not provide simple laws based on $d$, $T$, $\varepsilon$, $\phi$, etc. because they are performed for a peculiar system under some given thermodynamic and dynamical conditions.

Here, we address the problem of fluid diffusion in ultra-confining disordered nanoporous materials by reporting robust physical laws established in the framework of surface/pore diffusion intermittence. By mapping molecular dynamics (MD) simulations onto mesoscopic random walk (RW) calculations

accounting for surface residence, our hierarchical approach captures the fluid diffusion in disordered nanoporous media and their underlying complex diffusivity landscapes. Moreover, by varying the matrix porosity $\phi$ and pore size $d$ but also the fluid/solid interaction strength $\varepsilon$, the proposed approach provides a means to quantitatively bridge the microscopic and mesoscopic dynamics in such complex environments using simple parameters. Both the typical surface residence and relocation times – $t_A$, $t_B$ – are found to derive from physical laws involving the pore size $d$ and the fluid/solid interaction strength normalized to the thermal energy $\varepsilon/k_BT$. In more detail, $t_A$ is shown to obey a transition state theory $t_A \sim t_A^0 \exp(-\Delta F/k_BT)$ where $\Delta F \sim \varepsilon$ is the free energy barrier that must be overcome to escape from the interaction field generated by the solid and $1/t_A^0$ is the characteristic escape attempt frequency. $t_A^0$ is found to be of the order of $\sim 10^{-12}$ s (a commonly accepted value) with a correction that accounts for pore diameter $d/\xi$ (with $\xi \sim \sigma$, i.e. the molecule size). As for the relocation time $t_B$, it is shown to scale with $d$ as quantitatively predicted by introducing a time cutoff $t_c \sim d^2/D_0$ in the relocation first passage probability distribution.

## Results

Our coarse grain model is developed in the spirit of the continuous time random walk (CTRW) as first proposed by Montroll and Weiss[25] and later extended by Shlesinger and Klafter[26] to the Levy walk model and other variants. The intermittent dynamics proposed in our approach involves a waiting time distribution at the pore surface coupled to a bridge statistics taking into account the first passage probability to connect one point at the interface to another through a random walk in the accessible pore network. The latter statistics couples distance and time as in the Levy walk (coupled memory) which also pertains to the Knudsen regime[27]. In the present approach, we will mainly consider the time distribution of these bridge statistics which are mapped onto atom-scale dynamics simulations to establish a bridge between the microscopic and mesoscopic scales. While the mapping proposed in this paper is derived for a simple fluid confined in prototypical models of highly disordered materials, we believe it can be extended to a much broader class of fluid/solid couples. However, such generalization must be performed with caution as there are a number of limitations which can lead to departure from the simple intermittent Brownian motion at the heart of our approach. Depending on the nature of the confined fluid and host solid, different molecular interactions are at play which are either short (e.g. dispersion, repulsion) or long (e.g. electrostatic, polarization) ranged. While such molecular interactions often lead to similar confined diffusivity mechanisms, they can induce more complex behaviors that are not entirely captured by a simple stop-and-go process. Moreover, for host solids with spatially-extended pore correlations (e.g. fractal solids), additional complexity and/or additional specific effects are expected.

**Different topological porous media.** Fluid diffusion in disordered nanoporous media was investigated by considering a set of 13 heterogeneous carbon structures with different densities $\rho_s$, porosities $\phi$, and pore sizes $d$. These structures—referred to as $CS_x$ with $x$ the density $\rho_s$ ranging from 0.5 to 1.4 g/cm³—were created using a quenching procedure (see Methods for full details). Typically, using a cubic box of length 100 Å containing $\sim$ 25,000 to $\sim$ 70,000 carbon atoms depending on $\rho_s$, molecular dynamics in the NVT ensemble was used with the reactive empirical bond order potential[28] in LAMMPS[29] to allow for bond formation/breaking during a 5 ns quench from 3000 K to 300 K. As an example, Fig. 1a shows the sample $CS_{0.70}$ filled with fluid molecules at their boiling point (as described in the Methods section, the adsorbed fluid density was

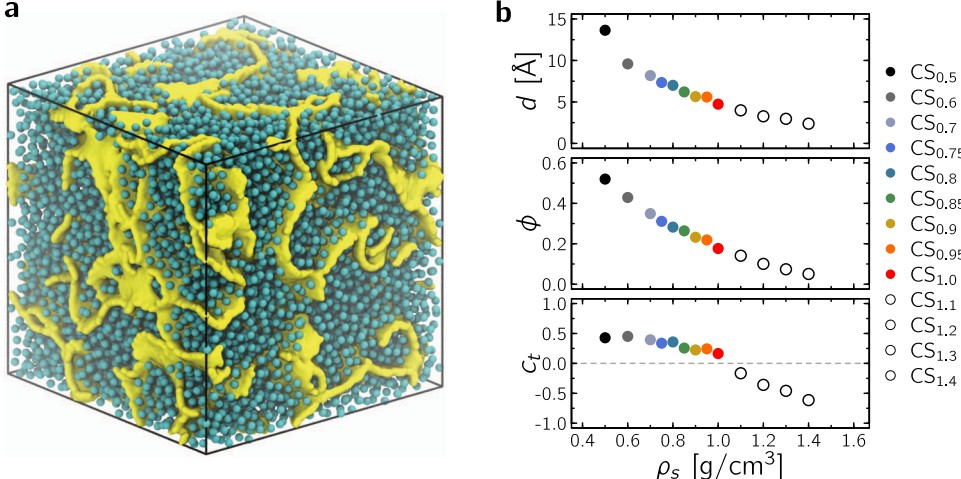

**Fig. 1 Prototypical disordered nanoporous structures. a** Molecular configuration for methane in a disordered nanoporous carbon (here $CS_{0.70}$, box length 100 Å). The cyan spheres are methane molecules while the yellow surface represents the carbon porous network. **b** Mean pore diameter $d$ (top), porosity $\phi$ (middle), and connection number $c_t$ (bottom) as a function of structure density $\rho_s$ of the digitized pore networks. The same color code applies throughout the manuscript. The samples obtained at larger densities—shown as white/open circles—possess unconnected porous networks ($c_t < 0$) so that no diffusion is observed in the long time limit.

estimated using standard Monte Carlo simulations in the Grand Canonical $\mu VT$ ensemble). Figure 1b shows the porosity $\phi$ as a function of $\rho_s$ where $\phi$ is determined using a Monte Carlo algorithm; by inserting $N$ probe molecules at random positions inside the simulation box containing the porous structure, the porosity can be estimated as the ratio $\phi \sim N_v/N$ (where $N_v$ is the number of probe molecules that do not overlap with any of the porous structure atoms). For each structure, provided a large number $N_v$ is considered, the pore size distribution $f(r)$ can be assessed from the diameter of the largest sphere containing each of the $N_v$ points (Supplementary Fig. 4)[30,31]. As expected, both the porosity $\phi$ and the mean pore size $d = \int rf(r)dr$ decrease upon increasing $\rho_s$ with $\phi \in [\sim 0.1, \sim 0.5]$ and $d$ varying from a few to ~15 Å.

Only structures with connected porosity were retained to investigate multiscale diffusion as unconnected porous samples necessarily yield zero self-diffusivities in the long time limit. For each sample, as described in the Methods section, the connectivity of the porous subspace accessible to a diffusing molecule was determined using the retraction graph associated with the digitized pore network (which conserves the topology at all scales). Such digitized binary sets are used to compute the connection number[32–34]:

$$c_t = -(\alpha_0 - \alpha_1)/\alpha_0 \tag{1}$$

where $\alpha_0$ and $\alpha_1$ are the number of vertexes (either isolated or connected) and links, respectively. $c_t$, which is a simple intensive parameter related to the number of irreducible paths per vertex, is invariant under any continuous pore network deformation. For structures with no isolated vertexes, the number of vertexes is smaller than the number of links, i.e., $\alpha_0 < \alpha_1$, so that $c_t > 0$ with a value that increases with pore network connectivity—in this case, the average number of links around a connected vertex is given by $<N_c> = 2(c_t + 1)$. On the other hand, for poorly connected pore networks, $\alpha_0 > \alpha_1$ so that $c_t < 0$ with $c_t \in [-1, 0]$—in this case, $<N_c> \rightarrow 0$ as $c_t \rightarrow -1$ so that the topological structure reduces to a set of isolated vertexes. In the above picture, the crossover $c_t = 0$ is generally assumed to correspond to a percolation threshold [32,33]. Figure 1b shows that $c_t > 0$ for $\rho_s \le 1.0$ g/cm³ as expected for a connected pore network (although $c_t$ is lower than typical values for very open networks $c_t \sim 0.5–0.7$[32,33]). On the other hand, $c_t < 0$ for $\rho_s > 1.0$ g/cm³, therefore indicating a long-range network disconnection for these dense porous structures.

**Intermittent Brownian motion with underlying stop-and-go diffusion.** Diffusion in the disordered media with connected porosity ($c_t > 0$) was investigated using MD for a simple Lennard–Jones (LJ) fluid at constant temperature and for varying fluid/solid interaction strengths. For such subnanoporous materials with strongly disordered pore morphologies, provided the number of adsorbed/confined molecules is low enough, the self-diffusivity $D_s$ is close to the collective diffusivity $D_c$ as cross-terms between fluid molecules are negligible (because fluid-solid interactions largely prevail over fluid-fluid interactions)[16]. As a result, due to the formal equivalence between permeance and collective diffusivity, i.e. $K = D_c/\rho k_B T \sim D_s/\rho k_B T$, the self-diffusivity also provides key insights into transport mechanisms under flow conditions as induced by pressure/chemical potential gradients. The LJ fluid parameters ($\sigma_0 = 3.81$ Å, $\varepsilon_0/k_B = 148.1$ K) were chosen to match those for methane—a simple nearly spherical probe. An isotropic molecular model—known as the united atom model—was used to describe the methane molecule. Such a simplified model was selected as it simply corresponds to a Lennard–Jones potential that is representative of a broad class of atomic and molecular liquids. Despite this simple fluid hypothesis, we believe that our approach can be extended to more complex fluids such as dipolar molecules. In particular, even if complex molecular structures lead to richer surface thermodynamics behavior with strong adsorption in specific sites and/or relocation with large inherent activation energies, the present approach remains relevant as such complexity is embedded—at least in an effective fashion—into the mean relocation and residence times. For each disordered porous structure, different fluid/solid strengths were considered: $\varepsilon/k_B T = n$ with $n = 0.01, 0.1, 0.2, 0.3, 0.5, 0.8,$ and 1. Varying $\varepsilon$ drastically affects the porosity seen by the confined fluid since it also modifies the repulsive interaction contribution—thus inducing large changes in the effective diffusivity of the confined fluid. To probe fluid dynamics at constant porosity while scanning a broad range of $\varepsilon$, the fluid/surface LJ potential was modified using a smoothing procedure involving a sigmoid function to rescale the potential well-depth at nearly constant repulsive contribution (see Methods for details). The temperature was chosen equal to $T = 450$K $\sim 3\varepsilon_0/k_B$ to ensure that the Fickian regime is reached in all cases over the typical simulation run length (20 ns).

Supplementary Fig. 6 shows the mean square displacement $<|\mathbf{r}(t) - \mathbf{r}(0)|^2>$ as a function of time $t$ for methane confined in the different disordered nanoporous materials (only data for $\varepsilon/k_B T = 0.1$ are shown for clarity). Typically, for the disordered materials considered here, the Fickian regime is reached after a few ns as each molecule diffuses over a length scale of the order of the simulation box size $L \sim 10$ nm. While such convergence is reached within typical timescales probed using molecular dynamics for these disordered materials with connected porosity ($c_t > 0$), there are materials classes where the long-time limit extends to much longer timescales. This includes solids with long-range pore correlations such as in fractal media or strong persistence length such as in one-dimensional pores. As shown in the inset in Fig. 2a, for all systems, the self-diffusivity $D_s$ – which is inferred from the Fickian regime in the long time limit $D_s = \lim_{t\to\infty} <|\mathbf{r}(t) - \mathbf{r}(0)|^2>/6t$ – is lower than the bulk self-diffusivity $D_s^0$. As a result, the tortuosity $\tau_{MD} = D_s^0/D_s$ – defined as the ratio of the bulk to the confined self-diffusivities—is larger than 1 as shown in Fig. 2a. As expected, upon increasing $\varepsilon/k_B T$, the average fluid/surface energy $\langle U_{fw}\rangle$ becomes more negative (attractive) so that $\tau_{MD}$ increases due to the increased tortuosity adsorption/residence contribution. Moreover, $\tau_{MD}$ increases upon increasing the solid density $\rho_s$ as more severe confinement leads to smaller diffusivity (as shown in Fig. 1, the pore size $d \sim \rho_s^{-x}$ with $x \sim 1$). The underlying microscopic diffusion mechanism in such ultra-confining materials can be identified by computing the self-correlation function $G_s(r, t)$. In particular, in an isotropic medium, $4\pi r^2 G_s(r,t)dr$ is the probability distribution that a molecule moves by a distance $r$ over a time $t$. As shown in Fig. 2b (see black dashed line), upon averaging over all molecules and time origins, the mean square displacement $<|\mathbf{r}(t) - \mathbf{r}(0)|^2>$ displays a smooth behavior $\sqrt{<\Delta r(t)^2>} \sim \sqrt{t}$ from which a confined self-diffusivity can be derived. Yet, Fig. 2b reveals that $4\pi r^2 G_s(r, t)$ displays a complex behavior characteristic of stop-and-go processes where the molecules switch from one location to another through jumps (the data shown here correspond to the sample CS$_{0.70}$ but the same data can be found in Supplementary Fig. 8 for different samples and fluid/surface interaction strengths). In more detail, the probability distribution exhibits marked vertical stripes indicating that molecules tend to remain within the same spatial domain over a given time. The distance between two stripes, which roughly corresponds to the fluid molecule size $\sigma_0$, corresponds to the jump amplitude. The typical residence time at a given position is given by the decay along the $t$ axis. Such stop-and-go diffusion was already reported by Sahimi and coworkers[35] in molecular dynamics of gas diffusion in a carbon nanotube/polymer composite and, more recently, by Kulasinski et al. for water diffusion in amorphous hydrophilic systems[36].

To shed more light into the complex diffusivity landscape in such disordered porous media, a single trajectory $R(t) = \sqrt{(\mathbf{r}(t) - \mathbf{r}(0))^2}$ is provided as an example in Fig. 2b together with a visualization of the corresponding molecular trajectory in Fig. 2c (to interpret these different space-time domains, Supplementary Fig. 9 provides additional individual trajectories). Such individual trajectories are typical but not necessarily fully representative as they were chosen to identify well-defined steps. However, the mechanisms discussed below are common to all molecules and lead to the heterogeneous behavior observed in $G_s(r, t)$. Before going into details, we define here a cavity as a portion of the pore network of size $d$. The first narrow stripe corresponds to molecules located in a given site with displacements over short distances $r$ much smaller than the molecule

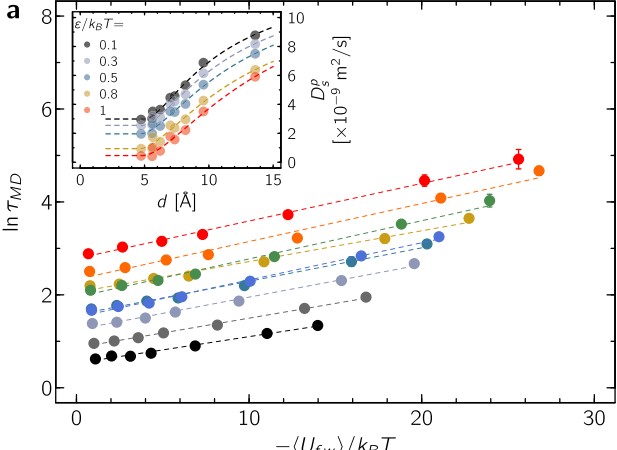

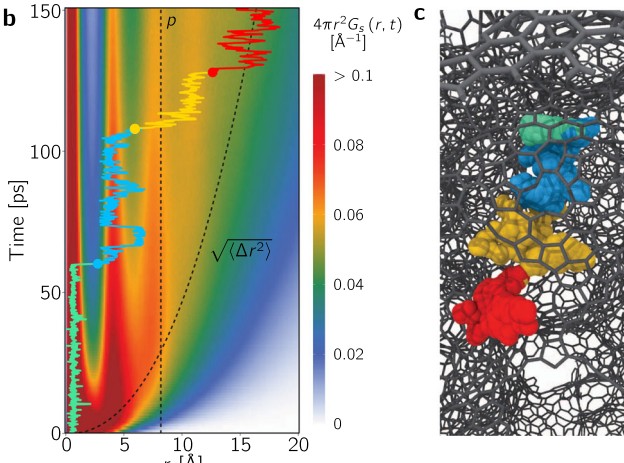

**Fig. 2 Complex stop-and-go fluid dynamics. a** Tortuosity $\tau_{MD} = D_s^0/D_s$ as a function of the average fluid/surface energy $<U_{fw}>/k_B T$ as assessed from MD for methane in a disordered nanoporous carbon (same color code as Fig. 1). The dashed lines are linear fits (note that the plot shows $\ln(\tau_{MD})$). The self-diffusion coefficients $D_s$ are obtained from the mean square displacements shown in Supplementary Fig. 6. The inset shows the in-pore diffusivity $D_s^p$ as a function of mean pore size $d$ for various fluid/surface interaction strengths $\varepsilon/k_B T$. The dashed lines are fits against Eq. (2). The different parameters of this effective fit are shown in Supplementary Fig. 7. The error bars correspond to the standard deviations around $\lim_{t\to\infty} <|\mathbf{r}(t) - \mathbf{r}(0)|^2>/6t$. **b** Contour plot showing the self-correlation function $4\pi r^2 \times G_s(r, t)$ as a function of $r$ and $t$ for methane in a disordered nanoporous carbon (here CS$_{0.70}$ with $\varepsilon/k_B T = 0.5$). The local color at a given $r, t$ position indicates $4\pi r^2 \times G_s(r, t)$ with the color scale shown on the right. The black dashed lines denote the mean pore size $d$ and mean displacement averaged over all molecules $\sqrt{<\Delta r(t)^2>} \sim \sqrt{<|\mathbf{r}(t) - \mathbf{r}(0)|^2>}$. The multicolor solid line shows a typical yet representative displacement $R(t)$ for a given molecule. Each color region (green, blue, orange, and red) indicates a portion of the 150 ps trajectory shown in (**c**) (see detailed description in the main text). For a more direct visualization, the corresponding trajectory is shown in (**c**) with successive molecular positions shown as spheres of the corresponding color. The carbon-carbon bonds in the porous structure are shown as gray sticks.

size $\sigma_0$. Such motions are illustrated by the green portion of the individual trajectory in Fig. 2b (in this specific example, the molecule is adsorbed in the vicinity of the host surface as shown in **c**). This analysis is confirmed by the fact that the typical residence time associated with this dynamical sequence increases

with increasing the fluid/surface interaction strength $\varepsilon$ (Supplementary Fig. 8). The second narrow stripe centered at about $r \sim 4$ Å ($\sim \sigma_0$) corresponds to molecules jumping to a neighboring site. As illustrated in the individual trajectory (blue portion), such displacements correspond to molecules relocating from an adsorbed site to another while remaining within the same cavity ($r < d$). The third narrow stripe centered at about $r \lesssim d$ corresponds to confined diffusion where molecules explore both the pore center and surface region but remain within the same cavity (as illustrated with the orange portion of the individual trajectory). Finally, upon further increasing the time, the displacement becomes larger than the pore size – $r > d$ – as the molecule is transferred from a cavity to another (as illustrated in the corresponding red portion of the individual trajectory shown in **c**). As expected, at large times/distances, typically when $r > d$, the probability distribution becomes more homogeneous as the detailed structural footprint of the disordered host matrix averages out into a single effective parameter corresponding to the tortuosity. In particular, in the long time limit, the dynamics reach a Fickian regime as the molecules diffuse over distances $r$ large enough compared to the pore size $d$.

Despite the intrinsic complexity of diffusion in such rough energy landscapes, the local, i.e. in pore, self-diffusivity can be derived formally using effective approaches[19,37,38]. By in-pore diffusivity, we refer here to the short time range where molecules remain within the same cavities while reaching a pseudo-Fickian diffusion regime (in other words, such transport coefficients at the pore scale do not include network effects such as tortuosity but contain the fingerprint of the pore geometry/morphology). In more detail, considering the mean-square displacements shown in Supplementary Fig. 6, $D_s^p$ can be assessed from the linear regime observed in the short time scale where $<|\mathbf{r}(t_d) - \mathbf{r}(0)|^2> \leq d^2$ (where $t_d$ is the time required to displace molecules over a distance equal to the pore size $d$). To further validate the inferred value, it was checked that it is consistent with the in-pore diffusivity estimated as $\sim d^2/6t_d$. The simplest effective framework consists of writing the effective pore-scale diffusivity $D_s^p$ as an average over the whole pore volume, $D_s^p = 1/N \times \int \rho(\mathbf{r})D_s(\mathbf{r})d\mathbf{r}$ where $\rho(\mathbf{r})$ and $D_s(\mathbf{r})$ are the local density and self-diffusivity at a position $\mathbf{r}$. Within the transition state theory, the bulk self-diffusion coefficient can be written as an activated process $D_s^0 \sim \exp[-\Delta F^0/k_B T]$ where $\Delta F^0$ is the activation free energy to set the molecules in motion. Here, we refer to the bulk phase taken at the same temperature but also the same density as the confined phase. Therefore, even if the bulk phase is a low-density gas (for which diffusion does not involve any activation energy), $D_s^0$ should be understood as the liquid-like diffusivity of the bulk fluid taken at the same liquid-like density. For a confined fluid, the activation energy for diffusion can be assumed to correspond to the bulk activation energy augmented by the fluid/surface potential $\zeta(\mathbf{r})$, $\Delta F = \Delta F^0 - \zeta(\mathbf{r})$ (the sign minus is due to the fact that the interaction potential is attractive and, hence, negative so that molecules are trapped in deeper energy sites with an escape time requiring a larger activation energy). With this assumption, $D_s(r) = D_s^0 \exp[\zeta(r)/k_B T]$[19]. For complex media, there is no simple expression for $\zeta(\mathbf{r})$ but we use here a simple form where $\zeta(\mathbf{r})$ is constant when the distance to the surface is smaller than $\sigma$ and decays exponentially beyond. Such a generic form leads to the following local self-diffusivity: $D_s(r) = D_s^s$ for $r > d/2 - \sigma$ while $D_s(r) = D_s^0 + (D_s^s - D_s^0) \exp[-(d/2 - \sigma - r)/r_0]$ for $r \leq d/2 - \sigma$ (where $D_s^s$ is the surface self-diffusivity in the vicinity of the pore surface while $D_s^0$ is the bulk, i.e. unconfined, self-diffusivity). This expression simply assumes that the self-diffusivity is equal to the surface diffusivity $D_s^s$ for distances within a critical size $\sigma$ from

the surface while it decays exponentially with a characteristic lengthscale $r_0$ towards the bulk diffusivity $D_s^0$ as the distance to the surface increases, as depicted on Supplementary Fig. 11. After a little algebra, assuming the pore density is homogeneous, i.e., $\rho(\mathbf{r}) \sim \rho$, one arrives at:

$$D_s^p(d) = \begin{cases} D_s^s & (\text{for } d < 2\sigma) \\ D_s^0 + 2/d \times (D_s^s - D_s^0)(\sigma + r_0 - r_0 \exp[-(d - 2\sigma)/2r_0]) & (\text{for } d \geq 2\sigma) \end{cases}$$

(2)

As shown in the inset of Fig. 2a, the above effective expression provides an accurate description of the simulated in-pore diffusivity $D_s^p$. Both the variations in pore size $d$ and fluid/surface interaction strength $\varepsilon$ are accurately captured. The parameters $D_s^s$, $D_s^0$, $r_0$ and $\sigma$ extracted from the fit against Eq. (2) can be found in Supplementary Fig. 7. As expected, the bulk self-diffusivity $D_s^0$ is found to be constant at a value $14 \pm 0.6 \times 10^{-9}$ m²/s. While the confined fluid density is an ill-defined quantity that depends on a given pore volume definition, we note that the bulk reduced density $\rho^* = \rho\sigma^3$ needed to match the bulk self-diffusivity $D_s^0 = 14 \pm 0.6 \times 10^{-9}$ m²/s inferred from this simple in-pore diffusivity model falls within the range [0.8–1] (see Supplementary Fig. 12 showing the self-diffusivity of bulk methane as a function of density at the temperature considered here). Recalling that the number of confined fluid molecules was obtained by filling each porous material at the fluid boiling point, such reduced densities further support the use of a simple effective model for the in-pore diffusivity as they correspond to typical liquid densities. Similarly, $\sigma$ is independent of $d$ and $\varepsilon$ with a constant value of $2.4 \pm 0.1$ Å so that the critical distance $\sigma$ for surface diffusion roughly corresponds to the fluid molecular size. Interestingly, the quality of this effective in-pore diffusivity model shows that the surface diffusion $D_s^s$ and scaling $r_0$ can be treated as constant parameters for a given $\varepsilon$. On the other hand, as expected, $D_s^s$ is found to decrease upon increasing $\varepsilon$ while $r_0$ increases upon increasing $\varepsilon$. Typically, upon varying $\varepsilon/k_B T$ from 0.1 to 1.0, $D_s^s$ decreases from 3 to $0.5 \times 10^{-9}$ m²/s while $r_0$ increases from 0.8 to 2 Å. The fact that the scaling parameter $r_0$ depends on $\varepsilon$ can be rationalized as follows. Even if the surface/fluid interaction potential decay is independent of $\varepsilon$, it generates a free energy landscape $\zeta(r)$ that includes many body—fluid/fluid and fluid/wall—effects which lead to an effective scaling $r_0$ that depends on $\varepsilon$.

While the combination rule above provides a quantitative description of the molecular dynamics data, it remains mostly effective as it relies arbitrary choices combined with an empirical description of the diffusivity landscape explored by the fluid molecules. First, $\zeta(r)$ should be seen as an effective free energy field that modulates the bulk self-diffusivity by accounting for local intermolecular interactions but also for local density/packing effects. Therefore, even with simple pore geometries, instead of a robust free energy field rigorously derived from intermolecular interactions, $\zeta(r)$ is an effective function which is used to describe the self-diffusivity decay upon increasing the distance to the pore surface. The constant surface diffusivity at the pore surface is used to account for the fact that adsorbed molecules explore homogeneously the surface region $\sim 2\sigma$. Moreover, even if the conclusions above are qualitatively independent of the different assumptions involved, the decomposition into surface and bulk-like diffusions is also sensitive to the exact scaling defined in Eq. (2) and the parameter $2\sigma$ used to define the surface layer. In particular, other efficient decomposition rules have been proposed such as a simple weighted sum of surface and volume diffusivities which was found to accurately describe the dynamics of water in nanoconfinement[39]. Moreover, such surface/volume partition and the resulting predictions in terms

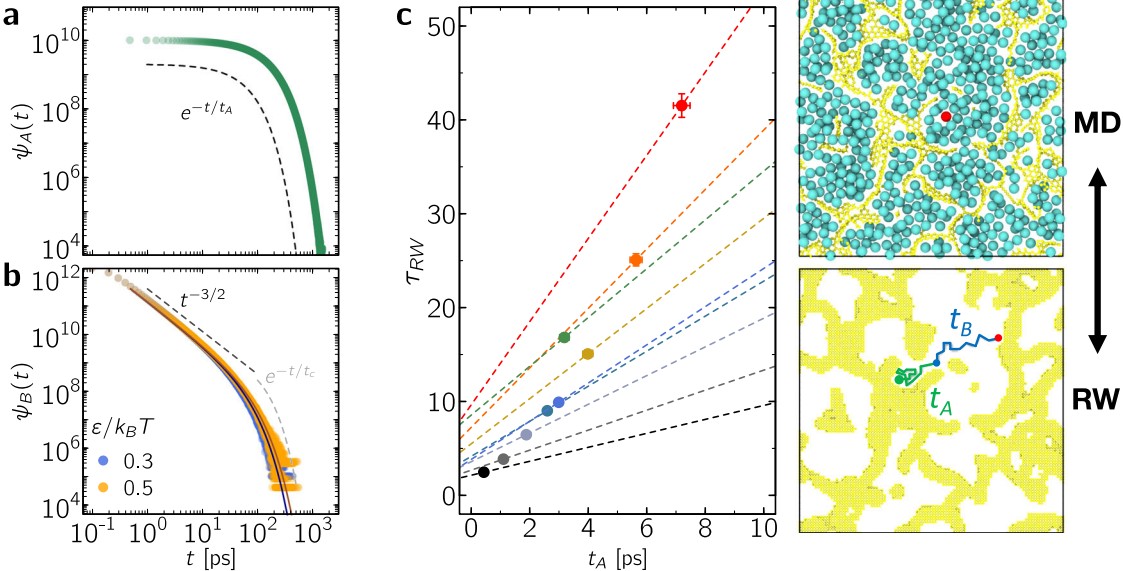

**Fig. 3 Mapping microscopic dynamics onto mesoscopic intermittent brownian motion. (a)** Typical residence statistics $\psi_A(t)$ with $t_A = 0.1$ ns. The dashed line shows the expected scaling $\psi_A(t) \propto e^{-t/t_A}$. **(b)** Bridge statistics $\psi_B(t)$ for $CS_{1.0}$ with two fluid/surface interactions $\varepsilon/k_B T$ as indicated in the graph. The solid lines are fits against Eq. (4), with $t_c = 42 \pm 0.5$ ps for $\varepsilon/k_B T = 0.3$ and $t_c = 51 \pm 0.7$ ps for $\varepsilon/k_B T = 0.5$. The dashed lines indicate the main regimes: $\psi_B(t) \propto t^{-3/2}$ and $\psi_B(t) \propto e^{-t/t_c}$. **(c)** Random walk tortuosity $\tau_{RW}$ as a function of the residence time $t_A$ (dashed lines). Only data obtained for $\varepsilon/k_B T = 0.5$ are shown here for the sake of clarity, but data for other surface/fluid interaction strengths show similar behavior. The data points are the projection of the molecular dynamics tortuosities $\tau_{MD}$ onto the Random walk tortuosities $\tau_{RW}$ to obtain the residence ($t_A$) and relocation ($t_B$) times; i.e. $\tau_{MD} \sim \tau_{RW}^0 [1 + t_A/t_B]$. The schematics on the right illustrate this mapping. In MD simulations, the self-diffusivity is probed by measuring the mean square displacement of the fluid molecules (here, a single molecule is marked in red but the self-diffusivity is averaged over each individual molecule). In RW calculations, the self-diffusion of particles is probed using a specific algorithm which physically accounts for the adsorption and relocation times $t_A$ and $t_B$ as illustrated in the corresponding schematic. The vertical error bars are the same as in Fig. 2 while the horizontal error bars are their projection onto the dashed lines.

of in-pore diffusivities $D_s^p$ are also dependent on the geometry choice—usually far from any realistic description—made to describe the pores in such disordered materials (planar, cylindrical or spherical). In practice, as will be shown in the rest of this paper, to avoid relying on such effective frameworks, the intermittent Brownian formalism mapped onto molecular dynamics data provides a means to describe stop-and-go processes in such disordered and ultraconfining materials without invoking any definition for the surface layer and the self-diffusion decay as molecules get closer to the pore surface.

The stop-and-go, i.e. intermittent, diffusion observed in our atom-scale dynamics simulations suggests that the corresponding data can be analyzed using the framework of intermittent Brownian dynamics. Indeed, as shown in Fig. 2b, while ensemble averaging over each molecule leads to a Fickian regime with an effective self-diffusivity, each individual trajectory involves intermittent motion with alternate series of in-pore diffusion and surface adsorption. In more detail, within this formalism, the mesoscopic, i.e., coarse-grained, dynamics beyond molecular time and length scales is governed by two parameters: the residence time $t_A$ during which a molecule remains adsorbed to the surface and the relocation time $t_B$ between two adsorption periods[10,14]. To probe such intermittent dynamics, the pore space $\Omega$ available for the dynamics of spherical molecules inside the carbon matrix was extracted by mapping a 3D lattice network having a voxel size $\Delta = 0.2$ Å (as explained in the Methods section). A voxel belongs to the pore space if its distance $x$ to any carbon center is $x > \sigma$ where $\sigma$ is the LJ parameter for the fluid/surface interaction. The surface boundary $\partial\Omega$ of $\Omega$ is made of surface voxels which are at the frontier between $\Omega$ and its complementary space. This allows defining a continuous space for molecular diffusion limited by the surface boundary. With the aim to simulate long-range intermittent dynamics, only the

greatest connected part $\Omega_c$ of $\Omega$ is considered (in the present study, for all samples $c_t > 0$, $\Omega_c$ percolates through the periodic minimal image). Intermittent Brownian motion was then simulated using the following advanced random walk approach. An interfacial volume is defined as $\partial\Omega_c \times x_0$ where $x_0 = 0.2$ pm is an infinitesimal thickness. Diffusion in the pore cavities is described using regular random walk simulations with a bulk-like self-diffusivity $D_s^p$ estimated from molecular dynamics. When a molecule center of mass reaches $\partial\Omega_c \times x_0$, it remains stopped for a time $t_S$ distributed according to an exponential probability density function having a first moment $t_A$. After $t_S$, the center of mass is placed at the distance $x_0$ from $\partial\Omega_c$ for a new relocation step. The procedure above leads to intermittent Brownian motion where the residence and relocation steps are distributed according to two underlying probability density functions $\psi_A(t)$ and $\psi_B(t)$ (having $t_A$ and $t_B$ as first moments). On the one hand, as illustrated in Fig. 3a, the residence times obey a statistics given by:

$$\psi_A(t) = 1/t_A \times \exp(-t/t_A) \qquad (3)$$

where $\psi_A(t)dt$ is the probability that the residence lasts a time between $t$ and $t + dt$. While the exponential decay in Eq. (3) provides a generic description of the residence time distribution, power-law distributions can be observed in other specific situations such as in media with surface heterogeneity or complex surface dynamics. However, as will be illustrated below, among possible behaviors, the exponential decay is important as it corresponds to a well-defined underlying thermodynamic picture where desorption corresponds to an activated mechanism. Moreover, considering the mapping between microscopic and mesoscopic tortuosities proposed in what follows, it only relies on the mean residence time and not the exact time distribution. On the other hand, $\psi_B(t)$ is the bridge statistics which describes the time distribution between a desorption event and the next first re-encounter within the

proximal zone $\partial\Omega_c \times x_0$. Such generic bridge statistics in confinement is illustrated in Fig. 3b which shows $\psi_B(t)$ for methane confined in the disordered sample $CS_{1.0}$ with different fluid/surface interactions. On the one hand, after a plateau in the very short time range ($\lesssim$ps), $\psi_B(t)$ decays as a power law $\psi_B(t) \sim t^{-3/2}$. On the other hand, in the long time regime, $\psi_B(t) \propto \exp(-t/t_c)$ as strong confinement in the sample cavities introduces a time cutoff $t_c$ in the relocation process since every confined molecule eventually returns to the surface within a finite time. This generic behavior for such a finite i.e. confining medium can be described as[40]:

$$\psi_B(t) \propto \psi_B^\infty(t) \exp[-t/t_c] \tag{4}$$

where $\psi_B^\infty(t)$ corresponds to the bridge statistics for a semi-infinite medium (denoted by the symbol $\infty$). As shown in Supplementary Notes, $\psi_B^\infty(t)$ can be determined by considering the trajectory of a molecule starting at a distance $x_0$ from the adsorbing region located in $x = 0$ and crossing this interface for the first time at a time $t$[41]:

$$\psi_B^\infty(t) = \frac{x_0}{\sqrt{4\pi D_s^p t^3}} \exp\left(-\frac{x_0^2}{4 D_s^p t}\right) \underset{t \gg x_0^2/D_s^p}{\sim} \frac{x_0}{\sqrt{4\pi D_s^p t^3}} \tag{5}$$

In this equation, the second equality corresponds to the solution in the limit $t \gg x_0^2/D_s^p$. Such expressions are valid for a semi-infinite medium where the probability to return to the surface becomes vanishingly small in the long time limit.

Figure 3c shows the tortuosity $\tau_{RW}$ as a function of the residence time $t_A$ as obtained using random walk simulations for the different $CS_x$ samples (only data for $\varepsilon/k_BT = 0.5$ are shown here for the sake of clarity). The dashed lines in Fig. 3c are RW results obtained by varying $t_A$ in a quasi-continuous manner. As expected, the tortuosity can be rescaled as:

$$\tau_{RW} = \tau_{RW}^0 \left(1 + \frac{t_A}{t_B}\right) \tag{6}$$

where $\tau_{RW}^0$ and $t_B$ only depend on the specific $CS_x$ sample considered. While $t_B$ is the typical relocation time, $\tau_{RW}^0$

corresponds to the geometrical tortuosity obtained for a vanishing residence time ($t_A \to 0$). As shown in Fig. 3c, projecting the $\tau_{MD}$ values obtained by MD (points) onto the ones obtained by RW (lines), i.e. $\tau_{MD} = \tau_{RW}$, allows mapping the molecular and mesoscopic tortuosities. This provides a means to estimate for each sample $CS_x$ the residence ($t_A$) and relocation ($t_B$) times as a function of the fluid/surface interaction strength $\varepsilon/k_BT$ (values that cannot be assessed using MD for such complex disordered materials). In more detail, $t_A$ and $t_B$ are such that $\tau_{MD} = \tau_{RW}^0(1 + t_A/t_B)$. Considering that $\tau_{RW}^0$ and $t_B$ for a given sample and $\varepsilon/k_BT$ are uniquely defined from the slope and intercept in Fig. 3c, there is only one set ($t_A, t_B$) that verifies $\tau_{MD} = \tau_{RW}$. As shown in our previous work[14], it should be emphasized that $t_A$ and $t_B$ can be directly estimated from molecular dynamics when simple pore geometries are considered. However, such calculations turn out to be extremely challenging for disordered porous media because the surface/volume decomposition is a complex ill-defined problem. Energy-based criteria such as surface-fluid energy cutoffs or geometrical criteria such as positions to the interface can be used but they rely on arbitrary choices. In contrast, the approach proposed in the present work provides a means to split the complex diffusivity behavior into residence and relocation steps without having to rely on these arbitrary choices.

**Bridging molecular/mesoscopic dynamics in disordered media.** The residence and relocation times are upscaled parameters which provide a mean to quantitatively bridge the microscopic and mesoscopic dynamics in porous media through the intermittent Brownian motion formalism. Yet, beyond simple mapping procedures like matching molecular and coarse-grained tortuosities, there is a need to establish robust and quantitative physical behaviors for $t_A$ and $t_B$. With this aim, the effect of mean pore size $d$ and fluid/surface interaction strength $\varepsilon/k_BT$ on $t_A$ and $t_B$ is shown in Fig. 4. In what follows, we first report a molecular

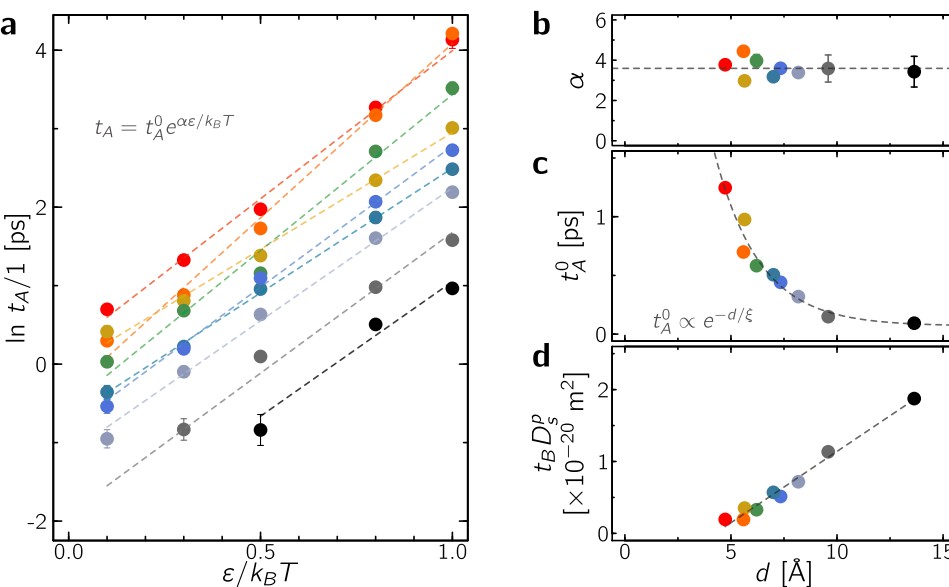

**Fig. 4 Residence and relocation times. a** Logarithm of the residence time $t_A$ as a function of $\varepsilon/k_BT$ where $t_A$ is obtained by mapping the molecular and mesoscopic tortuosities as shown in Fig. 3c. The dashed lines are fits in the form $t_A = t_A^0 \exp(\alpha\varepsilon/k_BT)$ with $t_A^0$ the characteristic residence time for vanishing fluid/surface interactions. The error bars are those given in Fig. 3c. **b** Impact of pore size $d$ on the dimensionless parameter $\alpha$; $\alpha = 3.6 \pm 0.4 \, \forall p$. The error bars correspond to fitting errors. **c** Characteristic residence time $t_A^0$ as a function of mean pore size $d$. The dashed line is a fit against an effective decaying function $t_A^0 \propto t_{A,\infty}^0 [1 + \gamma \exp(-d/\xi)]$ with the critical pore size $\xi = 2.1 \pm 0.5$ Å and $t_{A,\infty}^0 \sim 0.07$ ps. **d** Relocation time $t_B \times D_s^p$ as a function of the mean pore size $d$ for $\varepsilon/k_BT = 1$. As expected, $t_B \propto d/D_s^p$ as the finite time spent upon relocation in the cavities introduces an upper bound in the survival probability (see text).

model for the residence time $t_A$ and then discuss the behavior of the relocation time $t_B$ using the formalism of first passage processes.

*Residence time* $t_A$. Figure 4a suggests that $t_A$ follows an activation law for all samples: $t_A = t_A^0 \exp[-\Delta F^*/k_B T]$ with $\Delta F^* = -\alpha\varepsilon$. In this transition state theory, $\Delta F^*$ corresponds to the free energy barrier that must be overcome by a fluid molecule to escape from the interaction field generated by the host surface. As for $1/t_A^0$, it corresponds to the frequency with which the molecule attempts to escape the free energy minimum where it is located. While the activated behavior observed for $t_A$ might appear as a surprising result, it can be rationalized through simple thermodynamic arguments. Let us consider a thermodynamic model where the molecule is either adsorbed in the vicinity of the pore surface or in the pore center. As a first-order approximation, it can be assumed that the free energy difference $\Delta F \sim N_s\varepsilon$ where $N_s$ is the number of surface atoms interacting with the fluid molecule. In other words, with this assumption, the free energy of an adsorbed molecule corresponds to the sum of the interaction energies with each neighboring surface atom while the entropy and fluid-fluid interaction contributions are treated as constant. Considering that $\Delta F^* = \delta\Delta F$ with $\delta \gtrsim 1$ (since the free energy barrier is necessarily larger than or equal to the free energy difference between the adsorbed/non-adsorbed physical states), the scaling in Fig. 4a indicates that $\alpha = N_s\delta$. As shown in Fig. 4b, for all samples (i.e. regardless of pore size $d$), $\alpha \sim 3.6$ which leads to $N_s \lesssim 3.6$. Such a value, which is independent of the considered structure, seems realistic as this corresponds to an underlying molecular picture where an adsorbed molecule interacts with $N_s \sim 3$ to $4$ structure atoms. To validate this interpretation, we calculated for all interaction strengths $\varepsilon/k_B T$ and porous materials $CS_x$, the radial distribution function $g(r)$ between host carbon atoms and methane molecules. The number of local carbon neighbors $N_c$ contributing to the free energy barrier involved in the escape time from surface residence was then estimated by integrating $g(r)$ up to the location corresponding to the Lennard–Jones potential minimum $r_{min} = 2^{1/6}\sigma$, i.e. $N_c = \int_0^{r_{min}} 4\pi r^2 g(r)\rho dr$. Considering all structures and interactions strengths, we found $<N_c> = 3.6 \pm 1$ which is consistent with the value obtained for $\alpha$ in Fig. 4b.

Figure 4c shows that the prefactor $t_A^0$, which corresponds to the characteristic timescale for activated molecular desorption from the surface, is of the order of $\sim 1$ ps—a classical value used in transition state theories and nucleation models in dense liquid states. More importantly, $t_A^0 \sim t_{A,\infty}^0[1 + \gamma \exp(-d/\xi)]$ where $t_{A,\infty}^0 \sim 0.07$ ps corresponds to the value for infinitely large pores (vanishing confinement). The typical decay length $\xi \sim 2.1$ Å is of the order of the molecule size $\sigma$, therefore indicating that the correction to the escape attempt time is related to the pore size $d$. This can be understood by the fact that, for a given free energy barrier $\Delta F^*$, strong confinement leads to increasing residence times due to the decrease in molecular paths leading to desorption.

*Relocation time* $t_B$. Figure 4d shows that the relocation time $t_B$ scales as $t_B \sim d/D_s^p$. This result is not completely intuitive as it departs from a straightforward estimate obtained using the pore diffusivity $D_s^p$ and diffusion domain $\sim d$, i.e. $t_B \sim d^2/D_s^p$. Yet, as described quantitatively in what follows, the scaling $t_B \sim d$ can be rationalized by accounting for the fact that the diffusion, i.e. relocation, time within the confining cavities has necessarily an upper bound (due to the finite pore size, each molecule eventually readsorbs to the surface). This constraint, which is at the root of the scaling $t_B \sim d$, can be quantitatively predicted by introducing a time cutoff $t_c$ in the relocation first passage probability $\psi_B(t)$. In addition

to $t_c$, we also introduce a short time cutoff $t_0$ as $\psi_B(t)$ is necessarily equal to zero for times shorter than the time $t_0$ needed for a molecule to travel the minimum bridge of extension $x_{min}$.

As derived in the Supplementary Notes, with the lower/upper time limits $t_0$ and $t_c$, $\psi_B(t)$ simply writes $\psi_B(t) = C \times \exp(-t/t_c)/t^{3/2}$ for $t \in [t_0, t_c]$ where $C = [2\exp(-t_0/t_c)/\sqrt{t_0} - 2\sqrt{\pi/t_c}\,\mathrm{erfc}(\sqrt{t_0/t_c})]^{-1}$ is obtained by writing the normalization condition $\int_0^\infty \psi_B(t)dt = 1$. The first passage distribution for relocation $\psi_B(t)$ allows estimating the mean relocation time $t_B$ as:

$$t_B = \int_0^\infty t\psi_B(t)dt \qquad (7)$$

As shown in Supplementary Notes, upon inserting $\psi_B(t) \sim \exp(-t/t_c) \times t^{-3/2}$ for $t > t_0$ (0 otherwise) into Eq. (7), it can be shown that: $t_B = C \times \sqrt{\pi t_c}\,\mathrm{erfc}(\sqrt{t_0/t_c})$. By writing that $t_0 \ll t_c$ (i.e. $C \sim \sqrt{t_0}/2$), this expression simplifies as:

$$t_B \sim \frac{\sqrt{\pi t_c t_0}}{2} - t_0 \sim \frac{x_{min}}{4D_s^p}[\sqrt{\pi}\beta d - 2x_{min}] \qquad (8)$$

$t_c$ is associated with a geometrical cut-off length $r_c$ which indicates the maximal extension of a bridge. $r_c$ is of the order of the pore diameter $d$ and can be written as $r_c = \beta d$, where $\beta \sim 1$ is related to the accessible in-pore horizon. Assuming Fickian diffusion upon relocation, we can write $t_0 \sim x_{min}^2/2D_s^p$ and $t_c \sim \beta^2 d^2/2D_s^p$. As shown in Fig. 4d, by assuming that $x_{min}$ is independent of the pore structure, Eq. (8) provides a reasonable description of the observed scaling $t_B \sim d$ with a negative intercept in $d = 0$. Yet, as detailed in Supplementary Notes, $x_{min}$ can be estimated from the probability density function of the bridge displacement $\theta(r)$ where $r$ the is the end-to-end Euclidean distance of a Brownian bridge[42] [see Supplementary Fig. 10a]. With this refined analysis, as shown in Supplementary Fig. 10b, $x_{min}$ does depend on the pore diameter $d$. Taking into account this dependence, the simulated data $t_B \times D_s^p$ in Fig. 4b as a function of $d$ can be retrieved using a unique value $\beta \sim 0.7$ for all values $\varepsilon/k_B T$, as shown in Supplementary Fig. 10c.

## Discussion

The statistical physics approach reported in this paper provides an efficient mean to upscale microscopic dynamics in complex porous media to the engineering, i.e., continuum, level. This general and versatile method consists of upscaling molecular constants—typically, the adsorption strength and self-diffusivity—as obtained using molecular dynamics through the formalism of intermittent Brownian motion. While this robust framework is well-established for ordered materials with regular pore geometry and simple pore network topology, the present work extends its scope to ultra-confining disordered porous media with underlying complex free-energy landscapes. In particular, despite the complex interfacial dynamics in media involving ill-defined surface/volume regions, mapping of molecular dynamics simulations onto intermittent random walk provides a simple yet robust description through the mean surface residence ($t_A$) and in pore relocation ($t_B$) times. More importantly, using disordered porous materials with different porosities $\phi$/pore sizes $d$ but also fluid/surface interaction strengths $\varepsilon$, $t_A$ and $t_B$ are found to derive from basic physical models with parameters available to simple experiments. On the one hand, the mean residence time $t_A$ is simply related to the fluid/surface interaction strength $\varepsilon$ as it corresponds to the characteristic molecular escape time from a low (molecule in the surface vicinity) to a higher (bulk-like molecule in the pore center) free energy state separated by a free energy barrier $\Delta F^* \sim \varepsilon/k_B T$. On the other hand, $t_B$ can be simply predicted from the confined in-pore self-diffusivity $D_s^p$ and the corresponding mean-first

passage probability distribution which is truncated to account for the finite relocation time in confining cavities. Considering the mesoscopic, i.e., coarse-grained, description adopted in this approach, it is remarkable that all the problem complexity is embedded into two characteristic timescales that are related using simple physical laws to intrinsic material/fluid descriptors.

Such upscaling strategy could prove to be useful in numerous fields involving fluid adsorption and transport in porous materials: chemistry (e.g., adsorption, catalysis), chemical engineering (e.g., separation, chromatography), geosciences (e.g., pollutant transport), etc. In particular, among important examples relevant to such practical fields, the present approach can help describe molecular diffusion in the following applications: phase separation of gaseous or liquid effluents through porous media, filtration of small micro-pollutants such as organic/biomolecules, metallic and ionic complexes in water remediation, kinetics of products, reactants and by-products in catalytic processes, etc. From a practical viewpoint, conducting the exact upscaling strategy as reported in this paper can be quite involved; it requires building realistic porous material models and conducting both atom-scale and mesoscopic random walk simulations. However, the physical behavior of $t_A$ and $t_B$ as established above provides simple rules to predict the long-time fluid diffusion within a given porous material. In practice, all parameters needed to predict this macroscopic behavior are easily accessible experimentally; this includes the pore size $d$, the fluid/surface energy $\varepsilon$, and the self-diffusivity $D_s^p$. While $d$ can be estimated using adsorption-based techniques or derived using structural data, the fluid/surface energy can be probed from calorimetry or simply estimated from data for similar fluid/solid couples. As for $D_s^p$, a good approximation is to take this parameter equal to its bulk counterpart but more accurate data can be estimated by measuring the confined diffusivity using neutron scattering or NMR relaxometry. Inversely, starting from experimentally measured self-diffusivity in confinement, $t_A$ and $t_B$ can be extracted to shed light on physical phenomena occurring upon fluid adsorption, catalysis, etc. in a given porous material. In this context, our strategy can be coupled with free energy landscape computation to estimate the residence and relocation times. Such calculations are suitable for regular porous materials such as zeolites or metal-organic frameworks (for which dealing with a small porous subspace is sufficient thanks to symmetry considerations). However, such free energy approaches are nearly impossible for disordered porous materials with large representative elementary volume so that an effective approach based on simple physical laws is sound and robust.

Beyond regular adsorption/diffusion processes, our upscaling approach can be used to predict long-time effective diffusivity in problems involving more complex phenomena as observed in natural or anthropic disordered materials (wood, cement, etc.). This includes fluid/solid systems in which desorption is an activated process[43] but also processes involving reactive transport[44,45] and poromechanical effects such as adsorption-induced swelling[46]. Finally, the present approach can be used to obtain the elementary bricks to be implemented in mesoscopic numerical techniques such as finite elements calculations, pore network models[47], Lattice Boltzmann simulations but also more formal statistical physics approaches[20,48–50]. As already stated, our mapping procedure is expected to apply to a broad class of fluid/solid couples but some possible limitations must be considered as they can lead to more complex behaviors. Such limitations include the possible role of rich molecular interactions that are potentially long-ranged (e.g. electrostatic). Complex host solids with long-range pore correlations (fractal, low dimension) can also lead to additional complexity. In particular, in extremely narrow pores, confinement induces specific mechanisms such as molecular sieving[51] or single file diffusion[18] that depart from the

Fickian regime considered here. Moreover, by considering only percolating matrices ($c_t > 0$), the present study does not address connectivity aspects which can lead to anomalous temperature behavior depending on the ratio of adsorption and connectivity effects[51].

## Methods

**Porous material models.** Different samples of densities ranging from 0.5 g/cm³ up to 1.4 g/cm³ were produced using the following method. For a given density $\rho_s$, the atoms are placed randomly in a cubic box of a size 100 Å (an H/C atomic ratio ~ 0.091 was selected as it corresponds to a typical, realistic value for such disordered porous carbons[52,53]). Starting from a large temperature, each molecular structure was quenched using molecular dynamics performed using the large-scale atomic/molecular massively parallel simulator (LAMMPS[29]). Molecular interactions were described using the reactive empirical bond order (REBO) potential[28] to allow for bond formation/breaking. The quenching procedure is performed in the NVT ensemble by continuously decreasing the temperature from 3000 K down to 300 K in the course of a 5 ns simulation run. Three representative structures are presented in Supplementary Fig. 1 and all .xyz structure files are available upon request.

**Grand canonical Monte Carlo.** We simulated methane adsorption isotherms at 111.7 K in the various host structures (Supplementary Fig. 2) using Grand Canonical Monte Carlo (GCMC) with the Lennard–Jones parameters gathered in Supplementary Table 2. The saturating vapor pressure of methane at this temperature is $P_0 = 101325$ Pa (boiling point). In GCMC simulations, we consider a system at constant volume $V$ (the host porous solid) in equilibrium with an infinite reservoir of molecules (methane) imposing its chemical potential $\mu$ and temperature $T$. For a given set $(T, \mu)$, the adsorbed amount is given by the ensemble average of the number of adsorbed molecules versus the pressure $P$ of the gas reservoir (the latter is obtained from the chemical potential according to the equation of state for the bulk gas). The adsorption isotherm is simulated by increasing or decreasing the chemical potential of the reservoir.

The skeleton is considered rigid and the energy $U^{\alpha\beta}(i,j)$ between the site $i$ of type $\alpha$ and the site $j$ of type $\beta$ is given by[54]:

$$U^{\alpha\beta}(i,j) = \sum_{i,j} 4\varepsilon_{ij}^{\alpha\beta} \left[ \left( \frac{\sigma_{ij}^{\alpha\beta}}{r_{ij}^{\alpha\beta}} \right)^{12} - \left( \frac{\sigma_{ij}^{\alpha\beta}}{r_{ij}^{\alpha\beta}} \right)^{6} \right] \qquad (9)$$

Equation (9) describes interactions through a 6–12 Lennard–Jones potential with parameters $\sigma_{ij}^{\alpha\beta}$ (size) and $\varepsilon_{ij}^{\alpha\beta}$ (energy). The Lennard–Jones parameters are reported in Supplementary Table 2 for the interactions between sites of the same type, the cross interactions being computed from the Lorentz–Berthelot rules:

$$\sigma^{\alpha\beta} = \frac{1}{2} \left( \sigma^{\alpha\alpha} + \sigma^{\beta\beta} \right) \qquad \varepsilon^{\alpha\beta} = \sqrt{\varepsilon^{\alpha\alpha}\varepsilon^{\beta\beta}} \qquad (10)$$

**Molecular dynamics.** The methane-saturated structures obtained by GCMC are then used as starting structures for molecular dynamics (MD) simulations. All MD simulations are performed with LAMMPS[29] using the lj/cut potential with the same same-site parameters as the ones used for the GCMC simulations. In all simulations, the porous solid is kept frozen while the probe molecules are simulated at a temperature of 450 K for a NVE production run of 20 ns after a NVT thermalization run of 500 ps. The integration time step is 1 fs and the configurations are saved every 1 ps. To assess the influence of fluid/surface interaction on the effective diffusivity—and, hence, the tortuosity—the fluid/surface interaction strength $\varepsilon$ was varied. In so doing, the repulsive interaction felt by the fluid molecules decreases upon decreasing $\varepsilon$ so that the porosity explored by the confined molecules increases (inset Supplementary Fig. 3). Consequently, due to this effect, the tortuosity for a given structure strongly evolves with $\varepsilon$ without being per se an effect of the interaction strength. To correct this effect, we developed a modified Lennard–Jones potential that keeps the repulsive contribution constant. This modified interaction potential uses a smooth sigmoid function $U(r)$ defined as:

$$U(r) = \frac{L_-(r)e^{sr_c} + L_+(r)e^{sr}}{e^{sr_c} + e^{sr}} \qquad (11)$$

where $s = 50$ and $r_c = 0.97\sigma$ are the slope and center of the sigmoid, respectively. $L_-(r)$ and $L_+(r)$ are the connected functions defined for $r < r_c$ and $r > r_c$. To keep the repulsive interaction constant, $L(r)$ was maintained fixed as:

$$L_{+/-}(r) = 4\varepsilon_{+/-} \left[ \left( \frac{\sigma}{r} \right)^{12} - \left( \frac{\sigma}{r} \right)^{6} \right] \qquad (12)$$

with $\varepsilon_- = k_B T$. As shown in Supplementary Fig. 3, upon varying $\varepsilon_+$, a modified Lennard–Jones potential with different fluid/surface interaction strengths can be defined while keeping the repulsive part constant.

**Topology characterization and diffusion pore space.** The pore space $\Omega$ available for the dynamics of the spherical methane molecule inside the carbon matrix was determined as follows. A 3D lattice network is first defined with a voxel size 0.02

nm. A voxel belongs to $\Omega$ if its distance to any carbon centers is above 3.605 Å (this value is used as it corresponds to the Lennard–Jones parameter $\sigma$ for the fluid/surface interaction). A voxel belonging to $\Omega$ is set to 1 (0 otherwise). Such 3D lattice network allows defining the surface boundary $\partial\Omega$ of $\Omega$ made of surface voxels at the border between $\Omega$ and its complementary space. This allows us to define a continuous space for molecular diffusion which is limited by the surface boundary. An interfacial volume is defined as $\partial\Omega_c \times x_0$ where $x_0$ is a thickness equal to 0.2 pm. Supplementary Fig. 5 illustrates this procedure by showing for the sample $CS_{1.0}$ the resulting digitized pore network and the corresponding retraction graph obtained with a porosity $\phi = 0.177$. The molecular trajectory can be described as an alternate succession of a surface adsorption step on $\partial\Omega_c \times x_0$ followed by a Brownian motion in the confined bulk $\Omega_c$ leading to a new relocation on the surface. The time step for the Brownian motion is set to 0.1 ps and the self-diffusion coefficient is estimated from molecular trajectories (mean square displacements) as obtained from molecular dynamics at very early time steps.

## Data availability

The data sets and molecular configurations generated during and/or analyzed during the current study are available from the corresponding authors upon request. All MD simulations were performed using the software LAMMPS (stable release from August 31st, 2018).

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

## Acknowledgements

This work was supported by the French Research Agency (ANR TAMTAM 15-CE08-0008 and ANR TWIST ANR-17-CE08-0003).

## Author contributions

C.B. built and characterized the models and performed the molecular dynamics simulations. P.L. performed the morphological/topological analysis of the samples and carried out the mesoscopic simulations. All authors analyzed the data and developed the theoretical model. B.C. wrote the manuscript with inputs from all authors.

## Competing interests

The authors declare no competing interests.
