## [Peer Review File · Nature Communications]

REVIEWER COMMENTS

Reviewer #1 (Remarks to the Author):

This article discusses diffusion in certain classes of disordered nanoporous materials via an intermittent surface/pore diffusion formalism to map molecular dynamics onto random walks. To assess the influence of fluid/surface interaction on the effective diffusivity, the fluid/surface interaction strength was varied. Some general scaling relationships were theoretically derived and shown to agree well with simulations.

The authors are well respected experts in statistical physics and the field of diffusion in disordered media, including anomalous diffusion. They present interesting results that are technically well analysed. Personally, I like the paper, but I have worked in this field for a long time. I appreciate the formal elegance of the approach. However, I think that the general reader could easily miss the impact and significance of this work, in the way it is described.

I would encourage the authors to better contextualise their work and, also, to describe their results in a language that is more accessible to a broad audience (this being Nature Communications and not, say, Journal of Chemical Physics, the Physical Review or even ACS Nano). An attempt to do so appears on p.16-17, but that is much too late and too little.

If I am slightly critical, I could say that this is “only” a study of a number of reconstructed, theoretical representations of carbons, without justification on why this class of materials with two parameters would be representative of the much broader range of (real) materials suggested by the title, or even the introduction. In fact, it is not. In addition, the fluids are not general; this does not have to be a problem, but it has to be stated. There are many types of interactions (including electrostatics, polarization, etc.) that are not included. Molecules could be complex, there could be ions, etc.

So, I think the authors need to clearly define, from the start (and repeated in the conclusions with opportunities for generalisation), what the class of materials is, and what type of materials it would reasonably correspond to in practice (which is certainly not all disordered materials).

Also, the language is very much that of statistical physics – this is fine for the technical sections, but terms need to be more carefully explained for the non-experts in that field. They use the term “fluid mechanics” a few times in a way that could be misunderstood by the general reader: pressure driven flow, convection and even turbulence might come to mind. I would stick to “diffusion”. A word like “propagator” is known to physicists, but it needs some explanation for others – even many readers interested in transport in porous media will not know what this is.

There are many excellent references, and most of them are well selected, both historical key papers and recent references, representative for the background of the presented research. However, I am surprised to see no reference at all to the work of Sahimi and Tsotsis, who did a lot of work on diffusion in disordered media, including carbons, and even including “artificial” carbons created in a related way. Beyond a simple reference, some comparison is in order.

Regarding the statistical methods to analyse diffusion, there is no question that this builds on a fantastic legacy by Pierre Levitz going back to the 1990s – the signature of that work on random walks in disordered media and anomalous diffusion is clear! Still, there is a close relation to some work by others as well, in particular the CTRW (continuous time random walk) formalism of Montroll and Weiss, going back to the 1960s. This has been further developed in the work of Barber and Ninham, Michael Shlesinger, Yossi Klafter and many others. This is not to just add references, but to contextualise the work. In fact, Coppens and Dammers, in a paper in Fluid Phase Equilibria in 2006, propose a similar picture of trapping/waiting times and flight times (here called relocation times) to describe diffusion in general heterogeneous nanoporous media, of a much broader range than here. This is not a claim that the presented work is a trivial extension of all of this, but it is about context, and describing this clearly, also highlighting the innovative steps and what more is learnt.

Other comments:

- The parameter "p" is used to describe an average pore radius, based on maximum inscribed spheres. This is a strange symbol to pick for a radius. I advise to change it, as it is easily confused with probability; at some point, including the abstract and on p.14, "p" is even called curvature, which is incorrect, as it is the inverse.
- p.2 refers to Ref. 9, which is about zeolites, which do not have a "simple pore morphology". The behaviour is very different in strong confinement vs. the weaker confinement of, say, Knudsen diffusion.
- On p.5, it is unclear what the authors mean with: "For such ultra-confining materials, the self-diffusivity D_s is close to the collective diffusivity D_c ". A zeolite is an ultra-confining material, and D_s is typically not equal to D_c . Even when the interactions are of the hard-sphere type, molecules cannot pass each other, which leads to negative correlations on the fluid motion; due to mutual exclusion, the centre-of-mass/collective diffusivity is not the same as the tracer/self-diffusivity.
- On p.7, it should be clarified when the Fickian regime is reached in the long-time limit, as this is the case for these materials, but it is not so when there are (very) long-range correlations in the medium, such as in fractal media, or persistence is very strong, as in single-file diffusion in one-dimensional channels.
- P.9: can the authors justify or rationalise the split of the self-diffusivity in a constant term and a term that decays exponentially from the wall? How does that correlate to the energy profile in the pore? If it holds, it should at least hold for a smooth cylindrical or a slit pore – is there evidence for this?
- P.10 "...brownian formalism ... provides a mean..." -> "...Brownian formalism ... provides a means..." (on p.12 also "provide a means" with an "s" at the end; same on p.16)
- Fig.3(a): the trapping/adsorption times follow an exponential dependence for this specific surface-fluid interaction potential, but this is not a general result. In fact, power law dependences (at least over a finite scaling range) are found for some disordered media, including those with chemical and geometrical (e.g., fractal) surface heterogeneity, and even due to surface dynamics (e.g., for water in protein channels). This again underscores that the requirements on the material and fluid properties must be better articulated. This ultimately affects Fig.4 as well, which is a very nice result, but again depends on the nature of the potentials!

Reviewer #2 (Remarks to the Author):

Dear editor, dear authors, allow me to begin by thanking you for giving me the opportunity to review this article which I found very interesting.

This study focuses on the description of gas diffusion in amorphous porous carbons. The objective is to propose a bottom-up approach that allows to upscale the physical mechanisms encountered at the molecular scale to the mesoscopic scale. Here, the authors do not propose a change of scale strictly speaking, but rather to establish a link between the results of molecular dynamics simulations, which are based on fundamental axioms, and random walker simulations, which require predefined physical laws as input. The comparison of the two approaches then makes it possible to verify the validity of these physical laws and to deduce the relevant parameters.

Convincingly, the approach highlights the intermittent nature of the diffusion of individual gas molecules in the porous network. The authors propose a model based on the definition of a characteristic residence time, linked to the immobilization of a molecule trapped by adsorption, and a characteristic relocation time, linked to molecular diffusion between two adsorption sites. The authors propose fairly simple physical models to quantify the statistics of these characteristic times and to relate them to physical parameters such as the average pore size, the size of the diffusing molecules as well as the energy of fluid/solid interaction by means of scaling laws.

The paper is well written, of high scientific quality, answers some scientific questions and asks others. Nevertheless, I think that the authors could elaborate further (at least in the supplementary information) on some crucial points of their methodology as well as on some aspects of the model that I think are overlooked.

Given the originality of the study, its scientific quality and its potential interest for a large audience, I recommend the publication of this article in Nature Communications after a few revisions related to the points discussed in details in the attached .pdf document.

Yours sincerely,
Romain Vermorel

Review for : Bridging scales in disordered porous media by mapping molecular dynamics onto intermittent Brownian motion

Dear editor, dear authors, allow me to begin by thanking you for giving me the opportunity to review this article which I found very interesting.

This study focuses on the description of gas diffusion in amorphous porous carbons. The objective is to propose a bottom-up approach that allows to upscale the physical mechanisms encountered at the molecular scale to the mesoscopic scale. Here, the authors do not propose a change of scale strictly speaking, but rather to establish a link between the results of molecular dynamics simulations, which are based on fundamental axioms, and random walker simulations, which require predefined physical laws as input. The comparison of the two approaches then makes it possible to verify the validity of these physical laws and to deduce the relevant parameters.

Convincingly, the approach highlights the intermittent nature of the diffusion of individual gas molecules in the porous network. The authors propose a model based on the definition of a characteristic residence time, linked to the immobilization of a molecule trapped by adsorption, and a characteristic relocation time, linked to molecular diffusion between two adsorption sites. The authors propose fairly simple physical models to quantify the statistics of these characteristic times and to relate them to physical parameters such as the average pore size, the size of the diffusing molecules as well as the energy of fluid/solid interaction by means of scaling laws.

The paper is well written, of high scientific quality, answers some scientific questions and asks others. Nevertheless, I think that the authors could elaborate further (at least in the supplementary information) on some crucial points of their methodology as well as on some aspects of the model that I think are overlooked.

Given the originality of the study, its scientific quality and its potential interest for a large audience, I recommend the publication of this article in Nature Communications after a few revisions related to the points discussed below.

Yours sincerely,

Romain Vermorel

General Remark

In their study, the authors analyzed gas diffusion in amorphous porous carbon models. Among the range of model materials generated by molecular simulations, they selected only those with percolating diffusion paths.

The criteria used to define percolation address the structure of the porous network. The latter is characterized by techniques for inserting probe molecules (here methane) with a criterion of non-overlap with the atoms of the solid phase. In other words, the analysis of the porous network topology is carried out at a temperature of 0 Kelvin. However, at a finite temperature, the probe molecules are able to partially overcome the repulsive forces exerted by the solid phase and this effect increases with temperature according to Arrhenius' law.

There must therefore exist a range of solid density for which the connection number c_t is sensitive to temperature, reflecting the fact that gas molecules may or may not have access to branches of the porous network whose connection is possible depending on their kinetic energy. I expect that some models, that are not percolating at 0 Kelvin, will become percolating at sufficiently high temperatures. This type of behavior would then result in an increasing permeance with temperature due to molecular sieving effects.

In this study, the amorphous carbon models studied do not exhibit these molecular sieving effects and their permeance is driven by adsorption effects. In this case, the permeance $K \sim D/\rho kT$ may decrease with temperature because the diffusion coefficient and the gas density in the pores are controlled by adsorption and thus exhibit similar activation energies. These effects have been highlighted in a paper by Botan et al.,¹ based on molecular simulations performed with amorphous carbon models very close to those used by the authors.

It seems to me essential that the authors discuss this point because molecular sieving effects are unavoidable in many applications involving amorphous porous carbons. However their model does not account for such effects.

Remarks and questions regarding the local diffusion model

- How do the authors calculate the value of the diffusion coefficient D_s^p from the molecular simulations (results shown in the inset of Figure 2(a)) ? This is not clear and does not seem trivial to me ! They should provide more information on this point.
- The expression of the local diffusion coefficient $D_s(r)$ given by the authors does not correspond to the form of the potential described in the text. The function $D_s(r)$ has an arbitrary form that can be justified, but if $D_s(r)$ is proportional to the Boltzmann factor of the potential $\phi(r)$, then the corresponding $\phi(r)$ does not decay according to a simple exponential law. The authors should revise this part of the manuscript.
- Authors should insist that the r position is taken with the center of the pore as a reference. I suggest they add a schematic in the supplementary information to illustrate this model.
- The authors use adjustable four-parameter fits to compare their model to the simulation results. Being personally reluctant to use more than two adjustable parameters in a fit, I would like the authors to give more details on the method used to fit their model

to simulation data.

- The bulk diffusion coefficient of methane can be directly calculated by molecular simulation (or even deduced from data sets published in the literature). Could the authors reduce the number of adjustable parameters by imposing the value of D_s^0 deduced from MD simulations of bulk methane carried out with their molecular model ?
- I don't understand why the r_0 parameter depends on ϵ . The range of solid/fluid interactions should not depend on this parameter but rather on σ (which is a constant) and the geometry of the pore (I think about curvature). Do the authors have an explanation for this ?
- If the surface diffusion coefficient coincides with the zone where the potential $\phi(r) = \phi_0$ is constant, then D_s^s should simply be written $D_s^s = D_s^0 \exp(\phi_0/kT)$. We can think that the depth of the potential well ϕ_0 is proportional to the interaction energy ϵ , which would make D_s^s dependent on ϵ according to an exponential law. Why do we observe a linear law in figure S7 (a) ? Why don't the authors directly fit a parameter ϕ_0 to compare it to ϵ rather than using the diffusion coefficient D_s^s ?

Remarks and questions regarding the random walk simulations

- It is my understanding that the authors impose the residence time statistics in their simulations, while the relocation time statistics emerges from the random walk simulations using D_s^p as an input from MD simulations. If this is correct, I think the authors should emphasize this point to clarify the approach.
- The authors compare the tortuosities obtained from RW and MD simulations to deduce the characteristic residence (t_A) and relocation (t_B) times. Isn't it possible to obtain

a direct estimate of these characteristic times by post-processing molecular simulation data ? It seems to me that the authors have the data for this, like those shown in figure 2b. This would allow to obtain the input parameter t_A for RW simulations and to compare the values of t_B obtained by the two approaches.

Remarks and questions on Bridging molecular/mesoscopic dynamics in disordered media

- The authors could compare the value of α (reported in Figure 4 b) to the average number of solid atoms encountered in the sphere of first neighbors of methane molecules by calculating the radial distribution function. This would allow them to quantitatively validate their interpretation of the result obtained for α .
- How do the authors explain that what they call the minimal relocation loop, x_{min} , is independent of the considered structure ?

Miscellaneous comments

- The stop and go brownian motion described in this study is similar to that of water in cellulose-like materials as reported in a paper by Kulasinski et al.² The authors might consider adding this reference to their list.
- This paper deals with gas diffusion in amorphous carbons. It does not seem to me that diffusion in a bulk gas is an activated process. What is the activation energy ΔF^0 of the bulk diffusion coefficient D_s^0 they are referring to ?
- $D_s(r) = D_s^0 \exp(\phi(r)/kT)$ because $D_s(r) < D_s^0$ when $\phi(r)$ is negative.
- The authors should use different notations for the porosity, ϕ , and the effective solid/fluid potential $\phi(r)$ found at page 9.

- The label and the caption in figure 4 (a) should state that $t_A = t_A^0 \exp(\alpha\epsilon/kT)$. Otherwise t_A would increase with temperature and decrease with ϵ .

References

- (1) Boğan, A.; Vermorel, R.; Ulm, F.-J.; Pellenq, R. J.-M. Molecular Simulations of Supercritical Fluid Permeation through Disordered Microporous Carbons. *Langmuir* **2013**, *29*, 9985–9990.
- (2) Kulasinski, K.; Guyer, R.; Derome, D.; Carmeliet, J. Water diffusion in amorphous hydrophilic systems: A stop and go process. *Langmuir* **2015**, *31*, 10843–10849.

Reviewer #3 (Remarks to the Author):

The present paper aims to use the intermittent surface/pore diffusion formalism to map molecular dynamics onto random walk in disordered nanoporous media. I have found particularly interesting the fact that the authors try to link systematically fluid dynamics concepts (e.g. surface residence, in-pore relocation, and their time constants) to underlying simple parameters (e.g. pore curvature and temperature-rescaled surface interaction). The paper is very interesting and definitely well written. Hence I definitively support the publication of this manuscript, after the following (optional) comments will be considered.

- 1) The authors insist on the fact that the decomposition into surface and bulk-like diffusions is sensitive to the exact scaling defined in Eq. (2). Moreover they found that the critical distance for surface diffusion roughly corresponds to the fluid molecular size. All these findings reminded me about another (volumetric) scaling which was proposed some years ago (doi: 10.1038/ncomms4565). I was wondering if the latter could help in quantifying the thickness of the surface region where diffusion is strongly affected by pore interactions.
- 2) The authors claim that the present approach provides a robust formalism to predict diffusion for any fluid in complex nanoporous media using fluid and material parameters available to simple experiments. It would be a significant benefit to report a simple example about practical implications in a real application.
- 3) I am not sure if the authors used an isotropic molecular model for the methane molecules in the MD simulations. If this is the case, I was wondering what would be the impact of molecule anisotropy on the surface residence and relocation times.

Reviewer #1

This article discusses diffusion in certain classes of disordered nanoporous materials via an intermittent surface/pore diffusion formalism to map molecular dynamics onto random walks. To assess the influence of fluid/surface interaction on the effective diffusivity, the fluid/surface interaction strength was varied. Some general scaling relationships were theoretically derived and shown to agree well with simulations.

General comments

General comment 1. *The authors are well respected experts in statistical physics and the field of diffusion in disordered media, including anomalous diffusion. They present interesting results that are technically well analysed. Personally, I like the paper, but I have worked in this field for a long time. I appreciate the formal elegance of the approach. However, I think that the general reader could easily miss the impact and significance of this work, in the way it is described. I would encourage the authors to better contextualise their work and, also, to describe their results in a language that is more accessible to a broad audience (this being Nature Communications and not, say, Journal of Chemical Physics, the Physical Review or even ACS Nano). An attempt to do so appears on p.16-17, but that is much too late and too little. If I am slightly critical, I could say that this is “only” a study of a number of reconstructed, theoretical representations of carbons, without justification on why this class of materials with two parameters would be representative of the much broader range of (real) materials suggested by the title, or even the introduction. In fact, it is not. In addition, the fluids are not general; this does not have to be a problem, but it has to be stated. There are many types of interactions (including electrostatics, polarization, etc.) that are not included. Molecules could be complex, there could be ions, etc. So, I think the authors need to clearly define, from the start (and repeated in the conclusions with opportunities for generalisation), what the class of materials is, and what type of materials it would reasonably correspond to in practice (which is certainly not all disordered materials).*

Reply 1. We agree with this important recommendation from the reviewer. In line with the reviewer’s comment, we believe that there are 3 important aspects to be considered carefully when estimating generalization/extension of our approach: (1) molecular interactions at play, (2) fluid nature and molecular structure, and (3) porous material morphology and topology. To address this important comment, we have added the two following discussions in the introduction and in the conclusion:

- Introduction (Page 3-4): “While the mapping proposed in this paper is derived for a simple fluid confined in prototypical models of highly disordered materials, we believe it can be extended to a much broader class of fluid/solid couples. However, such generalization must be performed with caution as there are limitations which can lead to departure from the simple intermittent brownian motion at the heart of our approach. Depending on the nature of the confined fluid and host solid, different molecular interactions are at play which are either short (e.g. dispersion, repulsion) or long (e.g. electrostatic, polarization) ranged. While such molecular interactions often lead to similar confined diffusivity mechanisms, they can induce more complex behaviors that are not entirely captured by a simple stop-and-go process. Moreover, for host solids

with spatially-extended pore correlations (e.g. fractal solids), additional complexity and/or additional specific effects are expected.”

- Conclusion (Page 20): “As already stated, our mapping procedure is expected to apply to a broad class of fluid/solid couples but some possible limitations must be considered as they can lead to more complex behaviors. Such limitations include the possible role of rich molecular interactions which are potentially long-ranged (e.g. electrostatic). Complex host solids with long-range pore correlations (fractal, low dimension) can also lead to additional complexity. In particular, in extremely narrow pores, confinement induces specific mechanisms such as molecular sieving \cite{botan_molecular_2013} or single file diffusion \cite{hahn_deviations_1998} that depart from the Fickian regime considered here. Moreover, by considering only percolating matrices ($c_t > 0$), the present study does not address connectivity aspects which can lead to anomalous temperature behavior depending on the ratio of adsorption and connectivity effects \cite{botan_molecular_2013}.”

General comment 2. Also, the language is very much that of statistical physics – this is fine for the technical sections, but terms need to be more carefully explained for the non-experts in that field. They use the term “fluid mechanics” a few times in a way that could be misunderstood by the general reader: pressure driven flow, convection and even turbulence might come to mind. I would stick to “diffusion”. A word like “propagator” is known to physicists, but it needs some explanation for others – even many readers interested in transport in porous media will not know what this is.

Reply 2. We agree with the reviewer. To address this comment, we have carefully read the manuscript to identify every word or sentence that can be confusing to the Nature Communications general readership. In more detail,

- The sentence evoking “propagators” on page 2 has been rephrased as: “... propagators - i.e. the probability that a molecule moves by a quantity r in a time t - are not necessarily Fickian”
- It has been verified that the term “fluid mechanics” is not used in our manuscript as we agree with the reviewer that this may be confusing for the general reader
- To avoid any misunderstanding, the expression “fluid dynamics” has been replaced by “fluid diffusion” in different places in the manuscript - see “fluid dynamics” highlighted in blue (3 times)

General comment 3. There are many excellent references, and most of them are well selected, both historical key papers and recent references, representative for the background of the presented research. However, I am surprised to see no reference at all to the work of Sahimi and Tsotsis, who did a lot of work on diffusion in disordered media, including carbons, and even including “artificial” carbons created in a related way. Beyond a simple reference, some comparison is in order.

Reply 3. We agree with the reviewer that references to such pioneering works were missing in our initial manuscript. To address this point, we have added in our revised manuscript references to the following contributions by Prof. Sahimi (Refs. 4 and 35 in the revised manuscript, page 2 and page 8):

[REF 4] M. Sahimi, *Flow and transport in porous media and fractured rock: from classical methods to modern approaches*. (Wiley & Sons, 2011).

[REF 35] S.Y. Lim, M. Sahimi, T.T Tsotsi and N Kim. *Molecular dynamics simulation of diffusion of gases in a carbon-nanotube-polymer composite*. *Phys. Rev. E* 76, 011810 (2007)

Moreover, following the reviewer's comment, we have also added the following discussion in our revised manuscript (Page 8): "Such stop-and-go diffusion was already reported by Sahimi and coworkers \cite{lim_molecular_2007} in molecular dynamics of gas diffusion in a carbon nanotube/polymer composite and, more recently, by Kulasinski *et al.* for water diffusion in amorphous hydrophilic systems \cite{kulasinski_water_2015}."

General comment 4. *Regarding the statistical methods to analyse diffusion, there is no question that this builds on a fantastic legacy by Pierre Levitz going back to the 1990s – the signature of that work on random walks in disordered media and anomalous diffusion is clear! Still, there is a close relation to some work by others as well, in particular the CTRW (continuous time random walk) formalism of Montroll and Weiss, going back to the 1960s. This has been further developed in the work of Barber and Ninham, Michael Shlesinger, Yossi Klafter and many others. This is not to just add references, but to contextualise the work. In fact, Coppens and Dammers, in a paper in *Fluid Phase Equilibria* in 2006, propose a similar picture of trapping/waiting times and flight times (here called relocation times) to describe diffusion in general heterogeneous nanoporous media, of a much broader range than here. This is not a claim that the presented work is a trivial extension of all of this, but it is about context, and describing this clearly, also highlighting the innovative steps and what more is learnt.*

Reply 4. We thank the reviewer for reminding us of these important contributions to the field. We agree with the reviewer that an additional discussion should be added to better introduce our contribution within the existing literature. To address this comment, we have added the following discussion in our revised manuscript (Pages 3-4): "Our coarse grain model is developed in the spirit of the Continuous Time Random Walk (CTRW) as first proposed by Montroll and Weiss \cite{montroll_random_1965} and later extended by Shlesinger and Klafter \cite{shlesinger_strange_1993} to the Levy walk model and other variants. The intermittent dynamics proposed in our approach involves a waiting time distribution at the pore surface coupled to a bridge statistics taking into account the first passage probability to connect one point at the interface to another through a random walk in the accessible pore network. The latter statistics couples distance and time as in the Levy walk (coupled memory) which also pertains to the Knudsen regime \cite{levitz_knudsen_1997}. In the present approach, we will mainly consider the time distribution of these bridge statistics which are mapped onto atom-scale dynamics simulations to establish a bridge between the microscopic and mesoscopic scales."

Moreover, following the reviewer's comment, we have also added the two following important references when discussing stop and go processes (page 2 and page 8):

[REF 11] M.-O. Coppens and A.J. Dammers, *Effects of heterogeneity on diffusion in nanopores-From inorganic material to protein and ion channels*. *Fluid Phase Equilibria* (2006) 241, 308-316.

[REF 36] K. Kulasinski, R. Guyer, D. Derome and J. Carmeliet. *Water diffusion in amorphous hydrophilic systems: A stop and go process*. (2015) *Langmuir* 31, 10843-10849

Other comments:

Comment 1. *The parameter “p” is used to describe an average pore radius, based on maximum inscribed spheres. This is a strange symbol to pick for a radius. I advise to change it, as it is easily confused with probability; at some point, including the abstract and on p.14, “p” is even called curvature, which is incorrect, as it is the inverse.*

Reply 1. We thank the referee for pointing this confusing notation. We have changed the symbol “p” to the symbol “d” (for *diameter*) throughout the article, supplementary information and figures. The sentence involving the term “curvature” was modified as follows in the abstract: “corrected for pore diameter *d* to account for curvature/confinement effects”. Similarly, the following sentence (page 17) “[...] therefore indicating that the correction to the escape attempt time scales with the local curvature $\sim p/\sigma$ ” was rewritten as: “[...] therefore indicating that the correction to the escape attempt time is related to the pore size *d*.”

Comment 2. *p.2 refers to Ref. 9, which is about zeolites, which do not have a “simple pore morphology”. The behaviour is very different in strong confinement vs. the weaker confinement of, say, Knudsen diffusion.*

Reply 2. We agree with the reviewer that this reference is misleading when cited in this context. Therefore, to address this comment, we have removed this citation when discussing diffusion in simple pore morphologies (however, the reference to this important review paper remains cited in other places in the manuscript).

Comment 3. *On p.5, it is unclear what the authors mean with: “For such ultra-confining materials, the self-diffusivity D_s is close to the collective diffusivity D_c ”. A zeolite is an ultra-confining material, and D_s is typically not equal to D_c . Even when the interactions are of the hard-sphere type, molecules cannot pass each other, which leads to negative correlations on the fluid motion; due to mutual exclusion, the centre-of-mass/collective diffusivity is not the same as the tracer/self- diffusivity.*

Reply 3. We agree with the reviewer that this statement is misleading as currently written. As shown in Ref. [Falk et al. Nature Communications 2015 - cited in our manuscript], the self and collective diffusivities in such ultra-confining environments are close to each other even in the limit of fully saturated cavities. However, we agree that this result is specific to very disordered materials (since zeolites show departure between the self and collective diffusivities). More importantly, rigorously, the cross terms responsible for the difference between self and collective diffusivities can only be negligible in the limit of vanishing adsorbed amounts. To address this comment, we have rephrased the following sentence to better reflect these hypotheses/statements (Page 6): “For such subnanoporous materials with strongly disordered pore morphologies, provided the number of adsorbed/confined molecules is low enough, the self-diffusivity D_s is close to the collective diffusivity D_c as cross-terms between fluid molecules are negligible (because fluid-solid interactions largely prevail over fluid-fluid interactions) \cite{falk_subcontinuum_2015}”.

Comment 4. *On p.7, it should be clarified when the Fickian regime is reached in the long-time limit, as this is the case for these materials, but it is not so when there are (very) long-range*

correlations in the medium, such as in fractal media, or persistence is very strong, as in single-file diffusion in one-dimensional channels.

Reply 4. We thank the reviewer for raising this important point that should be discussed in detail. In order to address this comment, we have added the following sentences in our revised manuscript (Page 7): “Typically, for the disordered materials considered here, the Fickian regime is reached after a few ns as each molecule diffuses over a length scale of the order of the simulation box size $L \sim 10$ nm. While such convergence is reached within typical timescales probed using molecular dynamics for these disordered materials with connected porosity ($c_t > 0$), there are material classes where the long-time limit extends to much longer timescales. This includes solids with long-range pore correlations such as in fractal media or strong persistence length such as in one-dimensional pores.”

Comment 5. *P.9: can the authors justify or rationalise the split of the self-diffusivity in a constant term and a term that decays exponentially from the wall? How does that correlate to the energy profile in the pore? If it holds, it should at least hold for a smooth cylindrical or a slit pore – is there evidence for this?*

Reply 5. This is another important point raised by the reviewer. First of all, we emphasize that $\varphi(r)$ [note that φ has been replaced by ζ in the revised version to avoid confusion with the porosity φ] should be seen as an effective potential - in fact, a free energy field - that modulates the bulk self-diffusivity by accounting for local intermolecular interactions but also local density/packing effects. Therefore, even with simple pore geometries, instead of a robust interaction field mathematically derived from intermolecular interactions, $\varphi(r)$ should be seen as an effective function which is used to describe the self-diffusivity decay upon increasing the distance to the pore surface. Typically, the fact that we consider a constant diffusivity in the pore surface arises from the fact that adsorbed molecules explore homogeneously the surface region $\sim 2\sigma$ - which justifies that we use a constant term for the surface diffusion coefficient. Second, and probably even more importantly, we recall that this part of the paper - which deals with the splitting of the self-diffusivity - is intended to illustrate that the complex in-pore diffusivity results from combined surface and bulk-like mechanisms. However, as correctly stated by the reviewer in his/her report, such splitting relies on several assumptions (including the weighing function used to decompose into surface and volume contributions but also the diffusivity scaling for each contribution) so that it should be regarded as effective only. This is what was meant in the initial manuscript: “*In practice, as will be shown in the rest of this paper, to avoid relying on such effective frameworks, the intermittent Brownian formalism mapped onto molecular dynamics data provides a means to describe stop-and-go processes in such disordered and ultraconfining materials without invoking any definition for the surface layer and the self-diffusion decay as molecules get closer to the pore surface.*” We understand from the reviewer’s comment that the motivations and limitations of the self-diffusivity decomposition into surface and volume contributions should be better discussed. To address this comment, we have added/rephrased the following discussion on pages 11-12: “While the combination rule above provides a quantitative description of the molecular dynamics data, it remains mostly effective as it relies on arbitrary choices combined with an empirical description of the diffusivity landscape explored by the fluid molecules. First, $\zeta(r)$ should be seen as an effective free energy field that modulates the bulk self-diffusivity by accounting for local intermolecular interactions but also for local density/packing effects. Therefore, even with

simple pore geometries, instead of a robust free energy field rigorously derived from intermolecular interactions, $\zeta(r)$ is an effective function which is used to describe the self-diffusivity decay upon increasing the distance to the pore surface. The constant surface diffusivity at the pore surface is used to account for the fact that adsorbed molecules explore homogeneously the surface region $\sim 2\sigma$. Moreover, even if the conclusions above are qualitatively independent of the different assumptions involved, the decomposition into surface and bulk-like diffusions is also sensitive to the exact scaling defined in Eq. (2) and the parameter 2σ used to define the surface layer. In particular, other efficient decomposition rules have been proposed such as a simple weighted sum of surface and volume diffusivities which was found to accurately describe the dynamics of water in nanoconfinement \cite{chiavazzo_scaling_2014}. Moreover, such surface/volume partition and the resulting predictions in terms of in-pore diffusivities D_s^p are also dependent on the geometry choice -- usually far from any realistic description -- made to describe the pores in such disordered materials (planar, cylindrical or spherical). In practice, as will be shown in the rest of this paper, to avoid relying on such effective frameworks, the intermittent Brownian formalism mapped onto molecular dynamics data provides a means to describe stop-and-go processes in such disordered and ultraconfining materials without invoking any definition for the surface layer and the self-diffusion decay as molecules get closer to the pore surface."

Comment 6. P.10 "...brownian formalism...provides a mean..." -> "...Brownian formalism ...provides a means..." (p.12 also "provide a means" with an "s" at the end; same on p.16)

Reply 6. We thank the referee for pointing out these typos which have been corrected in the revised manuscript.

Comment 7. Fig.3(a): the trapping/adsorption times follow an exponential dependence for this specific surface-fluid interaction potential, but this is not a general result. In fact, power law dependences (at least over a finite scaling range) are found for some disordered media, including those with chemical and geometrical (e.g., fractal) surface heterogeneity, and even due to surface dynamics (e.g., for water in protein channels). This again underscores that the requirements on the material and fluid properties must be better articulated. This ultimately affects Fig.4 as well, which is a very nice result, but again depends on the nature of the potentials!

Reply 7. We agree with the reviewer. For the relocation time t_B , we do see scaling as a power law in line with the reviewer's comment. In fact, the exponential cutoff seen in the long time limit does not stem from the fluid/solid interactions but from confinement as all molecules will eventually readsorb to the surface within an average finite time t_c (in other words, regardless of the fluid/solid molecular interactions, such exponential cutoff will always be observed). On the other hand, for the residence time t_A , we agree with the reviewer that other characteristic behaviors can be observed. To address this comment, we have added the following sentences in the revised manuscript (Page 13): "While the exponential decay in Eq. (3) provides a generic description of the residence time distribution, power law distributions can be observed in other specific situations such as in media with surface heterogeneity or complex surface dynamics. However, as will be illustrated below, among possible behaviors, the exponential decay is important as it corresponds to a well-defined underlying thermodynamic picture where desorption corresponds to an activated mechanism. Moreover, considering the mapping between microscopic and mesoscopic tortuosities proposed in what follows, it only relies on the mean residence time and not the exact time distribution."

Reviewer #2

Dear editor, dear authors, allow me to begin by thanking you for giving me the opportunity to review this article which I found very interesting. This study focuses on the description of gas diffusion in amorphous porous carbons. The objective is to propose a bottom-up approach that allows to upscale the physical mechanisms encountered at the molecular scale to the mesoscopic scale. Here, the authors do not propose a change of scale strictly speaking, but rather to establish a link between the results of molecular dynamics simulations, which are based on fundamental axioms, and random walker simulations, which require predefined physical laws as input. The comparison of the two approaches then makes it possible to verify the validity of these physical laws and to deduce the relevant parameters. Convincingly, the approach highlights the intermittent nature of the diffusion of individual gas molecules in the porous network. The authors propose a model based on the definition of a characteristic residence time, linked to the immobilization of a molecule trapped by adsorption, and a characteristic relocation time, linked to molecular diffusion between two adsorption sites. The authors propose fairly simple physical models to quantify the statistics of these characteristic times and to relate them to physical parameters such as the average pore size, the size of the diffusing molecules as well as the energy of fluid/solid interaction by means of scaling laws. The paper is well written, of high scientific quality, answers some scientific questions and asks others. Nevertheless, I think that the authors could elaborate further (at least in the supplementary information) on some crucial points of their methodology as well as on some aspects of the model that I think are overlooked. Given the originality of the study, its scientific quality and its potential interest for a large audience, I recommend the publication of this article in Nature Communications after a few revisions related to the points discussed in details in the attached .pdf document.

Yours sincerely, Romain Vermorel

General Remark

In their study, the authors analyzed gas diffusion in amorphous porous carbon models. Among the range of model materials generated by molecular simulations, they selected only those with percolating diffusion paths. The criteria used to define percolation address the structure of the porous network. The latter is characterized by techniques for inserting probe molecules (here methane) with a criterion of non-overlap with the atoms of the solid phase. In other words, the analysis of the porous network topology is carried out at a temperature of 0 Kelvin. However, at a finite temperature, the probe molecules are able to partially overcome the repulsive forces exerted by the solid phase and this effect increases with temperature according to Arrhenius' law.

There must therefore exist a range of solid density for which the connection number ct is sensitive to temperature, reflecting the fact that gas molecules may or may not have access to branches of the porous network whose connection is possible depending on their kinetic energy. I expect that some models, that are not percolating at 0 Kelvin, will become percolating at sufficiently high temperatures. This type of behavior would then result in an increasing permeance with temperature due to molecular sieving effects.

In this study, the amorphous carbon models studied do not exhibit these molecular sieving effects and their permeance is driven by adsorption effects. In this case, the permeance $K \sim$

D/pkT may decrease with temperature because the diffusion coefficient and the gas density in the pores are controlled by adsorption and thus exhibit similar activation energies. These effects have been highlighted in a paper by Botan et al., based on molecular simulations performed with amorphous carbon models very close to those used by the authors.

It seems to me essential that the authors discuss this point because molecular sieving effects are unavoidable in many applications involving amorphous porous carbons. However their model does not account for such effects.

Reply. We thank the reviewer for these interesting comments. We agree that connectivity aspects are not fully addressed in our paper since we only consider percolating matrices ($c_t > 0$). To address this comment, we have added the following sentences in the general discussion on the main limitations added to address Comment 1 by Reviewer 1 above (Page 20): “In particular, in extremely narrow pores, confinement induce specific mechanisms such as molecular sieving \cite{botan_molecular_2013} or single file diffusion \cite{hahn_deviations_1998} that depart from the Fickian regime considered here. Moreover, by considering only percolating matrices ($c_t > 0$), the present study does not address connectivity aspects which can lead to anomalous temperature behavior depending on the ratio of adsorption and connectivity effects \cite{botan_molecular_2013}.”

On the other hand, we are a little hesitant in adding a more detailed discussion on permeance effects that result from a competition between adsorption and connectivity/diffusion. Indeed, in the present paper, we do not specifically consider collective transport mechanisms such as permeability but only diffusivity. Therefore, we feel that adding such a discussion might be confusing for the general readership of Nature Communications. We hope that the reviewer will be satisfied with our added discussion which explicitly states the limitations of our model (including in terms of percolation/molecular sieving effects). Finally, as a last remark to this comment, we would like to emphasize that, for a given porous matrix, the parameter c_t is temperature-independent as it is only related to the matrix topology as described in the methods section (using retraction graphs). As a result, even if the porosity seen by the probe molecule (kinetic diameter σ) is involved in the determination of c_t , the value determined is temperature independent. In conclusion, while we agree with the reviewer that transport restrictions in poorly connected networks can always be overcome through diffusion at sufficiently high temperature, we believe that the parameter c_t is a unique, well-defined quantity that rigorously describes pore network connectivity.

Remarks and questions regarding the local diffusion model

Remark 1. *How do the authors calculate the value of the diffusion coefficient D_{sp} from the molecular simulations (results shown in the inset of Figure 2(a)) ? This is not clear and does not seem trivial to me! They should provide more information on this point.*

Reply 1. We thank the reviewer for raising this important question. We fully agree that this point was not clearly discussed in our initial submission. To address this comment, we have added/modified the following paragraph in our revised manuscript (Page 10): “Despite the intrinsic complexity of diffusion in such rough energy landscapes, the local, *i.e.* in pore, self-diffusivity can be derived formally using effective approaches \cite{bhatia_modeling_2008, schneider_transport_2016, chemmi_noninvasive_2016}. By in-pore diffusivity, we refer here to the short time range where molecules remain within the same cavities while reaching a

pseudo-Fickian diffusion regime (in other words, such transport coefficients at the pore scale do not include network effects such as tortuosity but contain the fingerprint of the pore geometry/morphology). In more detail, considering the mean-square displacements shown in Supplementary Fig. 6, D_s^p can be assessed from the linear regime observed in the short time scale where $\langle |r(t_d) - r(0)|^2 \rangle \sim d^2$ (where t_d is the time required to displace molecules over a distance equal to the pore size d). To further validate the inferred value, it was checked that it is consistent with the in-pore diffusivity estimated as $\sim d^2/6t_d$."

Remark 2. *The expression of the local diffusion coefficient $D_s(r)$ given by the authors does not correspond to the form of the potential described in the text. The function $D_s(r)$ has an arbitrary form that can be justified, but if $D_s(r)$ is proportional to the Boltzmann factor of the potential $\varphi(r)$, then the corresponding $\varphi(r)$ does not decay according to a simple exponential law. The authors should revise this part of the manuscript.*

Reply 2. This is another important point raised by the reviewer. In fact, this point is very similar to that raised by reviewer 1 in his/her comment 5 (see above). Therefore, for convenience, we recall here our detailed reply together with the changes made to the manuscript to address this comment. First of all, we emphasize that $\varphi(r)$ [note that φ has been replaced by ζ in the revised version to avoid confusion with the porosity φ] should be seen as an effective potential - in fact, a free energy variable - that modulates the bulk self-diffusivity by accounting for local intermolecular interactions but also local density/packing effects. Therefore, even with simple pore geometries, instead of a robust interaction field mathematically derived from intermolecular interactions, $\varphi(r)$ should be seen as an effective function which is used to describe the self-diffusivity decay upon increasing the distance to the pore surface. Typically, the fact that we consider a constant diffusivity in the pore surface arises from the fact that adsorbed molecules explore homogeneously the surface region $\sim 2\sigma$ - which justifies that we use a constant term for the surface diffusion coefficient. Second, and probably even more importantly, we recall that this part of the paper - which deals with the splitting of the self-diffusivity - is intended to illustrate that the complex in-pore diffusivity is the result of complex surface and bulk-like mechanisms. However, as correctly stated by the reviewer in his/her report, such splitting relies on several assumptions (including the weighing function used to decompose into surface and volume contributions but also the diffusivity scaling for each contribution) so that it should be regarded as effective only. This is what was meant in the initial manuscript: *"In practice, as will be shown in the rest of this paper, to avoid relying on such effective frameworks, the intermittent Brownian formalism mapped onto molecular dynamics data provides a means to describe stop-and-go processes in such disordered and ultraconfining materials without invoking any definition for the surface layer and the self-diffusion decay as molecules get closer to the pore surface."* We understand from the reviewer's comment that the motivations and limitations of the self-diffusivity decomposition into surface and volume contributions should be better discussed. To address this comment, we have added/rephrased the following discussion: *"While the combination rule above provides a quantitative description of the molecular dynamics data, it remains mostly effective as it relies on arbitrary choices combined with an empirical description of the diffusivity landscape explored by the fluid molecules. First, $\zeta(r)$ should be seen as an effective free energy field that modulates the bulk self-diffusivity by accounting for local intermolecular interactions but also for local density/packing effects. Therefore, even with simple pore geometries, instead of a robust free energy field rigorously derived from intermolecular*

interactions, $\zeta(r)$ is an effective function which is used to describe the self-diffusivity decay upon increasing the distance to the pore surface. The constant surface diffusivity at the pore surface is used to account for the fact that adsorbed molecules explore homogeneously the surface region $\sim 2\sigma$. Moreover, even if the conclusions above are qualitatively independent of the different assumptions involved, the decomposition into surface and bulk-like diffusions is also sensitive to the exact scaling defined in Eq. (2) and the parameter 2σ used to define the surface layer. In particular, other efficient decomposition rules have been proposed such as a simple weighted sum of surface and volume diffusivities which was found to accurately describe the dynamics of water in nanoconfinement \cite{chiavazzo_scaling_2014}. Moreover, such surface/volume partition and the resulting predictions in terms of in-pore diffusivities D_s^p are also dependent on the geometry choice -- usually far from any realistic description -- made to describe the pores in such disordered materials (planar, cylindrical or spherical). In practice, as will be shown in the rest of this paper, to avoid relying on such effective frameworks, the intermittent Brownian formalism mapped onto molecular dynamics data provides a means to describe stop-and-go processes in such disordered and ultraconfining materials without invoking any definition for the surface layer and the self-diffusion decay as molecules get closer to the pore surface."

Remark 3. Authors should insist that the r position is taken with the center of the pore as a reference. I suggest they add a schematic in the supplementary information to illustrate this model.

Reply 3. We agree with the reviewer's suggestion. We have added in the Supplementary Information the following figure:

FIG. S11: Schematic representation showing the simple effective model based on the local self-diffusivity $D_s(r)$ given in Eq. (2). For $r > d/2 - \sigma$, molecules are adsorbed at the surface with a constant self-diffusivity $D_s(r) = D_s^s$. For $r < d/2 - \sigma$, the molecules are located in the pore center with a local self-diffusivity $D_s(r)$ depending on r . In this approach, $r = 0$ is the pore center where the diffusivity tends to the bulk value D_s^0 provided the pores are large enough.

We have also added in the revised manuscript a reference to this figure where appropriate (page 10).

Remark 4. *The authors use adjustable four-parameter fits to compare their model to the simulation results. Being personally reluctant to use more than two adjustable parameters in a fit, I would like the authors to give more details on the method used to fit their model to simulation data.*

Reply 4. We agree with the reviewer about this comment. In particular, we agree with the reviewer that fits with more than one parameter should be considered with caution. To address the reviewer comment, we have added the following information in the revised version of the SI, page 4: “All data treatment, fits and figures were made using the open source language R \backslash cite{R}. The fit for $D_s^p(d)$ in the inset of Fig. 2a was performed by applying Eq. (2) to the simulated data using the `nls()` function in R. The `nls` function performs Non-Linear Least Squares fitting using the Gauss-Newton algorithm. The initial guesses used are $D_0 = 12\text{e-}9$ m^2/s , $D_s^s = \min[D_s]$, $\sigma = 2 \text{ \AA}$, and $r_0 = 1 \text{ \AA}$. The error bars on Fig S7 were obtained from the fit by leaving all four parameters free [the standard errors on the fitting parameters are output by the `nls()` function]. To assess the fit quality, several fits were performed by imposing the values of D_0 or σ ; we found that the fit parameters converge to the same values as those shown in Fig. S7. In the end, we chose to show Figs. 2a and S7 with four free parameters to show that the impact of $\varepsilon/k_B T$ of these parameters does not depend on our choice to impose a given parameter or another.”

Remark 5. *The bulk diffusion coefficient of methane can be directly calculated by molecular simulation (or even deduced from datasets published in the literature). Could the authors reduce the number of adjustable parameters by imposing the value of D_s^0 deduced from MD simulations of bulk methane carried out with their molecular model ?*

Reply 5. This is another interesting point. While determining the bulk self-diffusivity D_s^0 is a rather straightforward task, estimating the density at which it should be taken is complicated. In fact, the pore volume (or, equivalently, the confined fluid density) is an ill-defined quantity as it depends on the arbitrary choice made to define the wall position. Therefore, to answer the reviewer’s question, we measured the self-diffusivity of bulk methane as a function of density at the temperature considered in our study. As can be seen from the figure below, we found that the bulk reduced density $\rho^* = \rho\sigma^3$ needed to match the bulk self-diffusivity $D_s^0 \sim 14 \times 10^{-9} \text{ m}^2/\text{s}$ inferred from the simple model for in-pore diffusivity falls within the range [0.8-1]. Such values, which correspond to typical liquid densities (remembering that the number of confined fluid molecules was estimated by filling each porous material at the fluid boiling point), further supports the use of a simple effective model for the in-pore diffusivity.

FIG. S12: Self-diffusivity D_s^0 of bulk methane at $T = 450$ K as a function of the reduced density $\rho^* = N/\sigma^3/V$. The dashed line is a guide to the eye while the shaded area indicates $D_s^0 = 14 \pm \times 10^{-9}$ m²/s as found in Fig. S7.

To carefully address the reviewer’s question, we have added in the revised version of the SI file, the figure above (see Figure S12). Moreover, we have added the following discussion in the revised version of the manuscript (Page 11): “While the confined fluid density is an ill-defined quantity that depends on a given pore volume definition, we note that the bulk reduced density $\rho^* = \rho\sigma^3$ needed to match the bulk self-diffusivity $D_s^0 \sim 14 \times 10^{-9}$ m²/s inferred from this simple in-pore diffusivity model falls within the range [0.8-1] (see Fig. S12 showing the self-diffusivity of bulk methane as a function of density at the temperature considered here). Recalling that the number of confined fluid molecules was obtained by filling each porous material at the fluid boiling point, such reduced densities further support the use of a simple effective model for the in-pore diffusivity as they correspond to typical liquid densities.”

Remark 6. I don’t understand why the r_0 parameter depends on ϵ . The range of solid/fluid interactions should not depend on this parameter but rather on σ (which is a constant) and the geometry of the pore (I think about curvature). Do the authors have an explanation for this ?

Reply 6. This is an important question raised by the reviewer. Indeed, as a first approximation, one would expect the typical decay length r_0 of the local self-diffusivity to be independent of the energy parameter ϵ . However, as already discussed in our reply to comment 2 above, $\varphi(r)$ [note that φ has been replaced by ζ in the revised version to avoid confusion with the porosity φ] is an effective potential. It corresponds to a free energy field which modulates the bulk self-diffusivity by accounting for local intermolecular interactions but also local density/packing effects. Therefore, even if the scaling of the surface/fluid potential is independent of ϵ , it generates a free energy landscape $\varphi(r)$ that includes many body - fluid/fluid and fluid/wall - effects. We understand from the reviewer’s comment that this point should be further clarified in our revised manuscript. Therefore, in addition to the modifications made in our reply to

comment 2, we have added the following sentences in our revised manuscript (Page 11): “The fact that the scaling parameter r_0 depends on ε can be rationalized as follows. Even if the surface/fluid interaction potential decay is independent of ε , it generates a free energy landscape $\zeta(r)$ that includes many body - fluid/fluid and fluid/wall - effects which lead to an effective scaling r_0 that depends on ε .”

Remark 7. *If the surface diffusion coefficient coincides with the zone where the potential $\varphi(r) = \varphi_0$ is constant, then D_s should simply be written $D_s = D_{s0} \exp(\varphi_0/kT)$. We can think that the depth of the potential well φ_0 is proportional to the interaction energy ε , which would make D_s dependent on ε according to an exponential law. Why do we observe a linear law in figure S7 (a) ? Why don't the authors directly fit a parameter φ_0 to compare it to ε rather than using the diffusion coefficient D_s ?*

Reply 7. As described in our reply to the previous point, φ_0 does not correspond to an energy parameter but to a free energy parameter which includes both enthalpy (energy) and entropy (structure) contributions. Therefore, direct comparison between φ_0 and ε is not straightforward even if the latter significantly contributes to the former. Moreover, it is important to note that in ultraconfining pores such as those considered in the present work the limit where $\varphi \sim \varphi_0$ is never reached. This makes the direct extraction of φ_0 impossible - a task which is even more complicated as the present work deals with complex materials having both disordered morphology and topology.

Remarks and questions regarding the random walk simulations

Remark 1. *It is my understanding that the authors impose the residence time statistics in their simulations, while the relocation time statistics emerges from the random walk simulations using D_{sp} as an input from MD simulations. If this is correct, I think the authors should emphasize this point to clarify the approach.*

Reply 1. We thank the referee for highlighting this lack of clarity. The reviewer is correct in his understanding of the mapping shown in Fig. 3c. First, τ_{RW} is computed from RW simulations in which t_A is imposed at different values (Lines in Fig. 3c). Second, τ_{MD} (points in Fig. 3c) are projected onto these lines to get the corresponding t_A . The paragraph including Eq. (6) on Page 14 was rephrased to better describe our approach: “Figure 3(c) shows the tortuosity τ_{RW} as a function of the residence time t_A as obtained using random walk simulations for the different CS_x samples (only data for $\varepsilon/k_B T = 0.5$ are shown here for the sake of clarity). The dashed lines in Fig. 3c correspond to RW results obtained by varying t_A in a quasi-continuous manner. [...] As expected, the tortuosity can be rescaled as: [...] While t_B is the typical relocation time, τ_{RW}^0 corresponds to the geometrical tortuosity obtained for a vanishing residence time ($t_A \sim 0$). As shown in Fig. 3(c), projecting τ_{MD} obtained by MD (data points) onto the RW results (lines), *i.e.* $\tau_{MD} = \tau_{RW}$, allows mapping the molecular and mesoscopic tortuosities. This provides a means to estimate for each sample CS_x the residence (t_A) and relocation (t_B) times as a function of the fluid/surface interaction strength $\varepsilon/k_B T$ (values that cannot be assessed using MD for such complex disordered materials).”

Remark 2. *The authors compare the tortuosities obtained from RW and MD simulations to deduce the characteristic residence (t_A) and relocation (t_B) times. Isn't it possible to obtain a direct estimate of these characteristic times by post-processing molecular simulation data ? It*

seems to me that the authors have the data for this, like those shown in figure 2b. This would allow to obtain the input parameter t_A for RW simulations and to compare the values of t_B obtained by the two approaches.

Reply 2. This is an interesting point raised by the reviewer. As shown in our previous work (Levitz et al., Soft Matter 2013), when using simple pore geometries, the mapping between the residence/relocation times obtained from random walk and those assessed from molecular dynamics match very accurately - both hydrophilic and hydrophobic pores having a slit and cylindrical geometry were considered. However, while such mapping is easily performed for pores having a simple shape, it turns out to be extremely challenging for disordered porous media because the surface/volume decomposition is ill-defined. Energy-based criteria such as surface-fluid energy cutoff or geometrical criteria such as position to the interface can be used but they rely on arbitrary choices. In contrast, the approach proposed in the present work provides a means to split the complex diffusivity behavior into surface residence and relocation steps without having to rely on these arbitrary choices. We understand from the reviewer's comment that this point should be better discussed in our manuscript. Therefore, to address this comment, we have added the following discussion in our revised manuscript (Page 14): *As shown in our previous work (Levitz et al., Soft Matter 2013), it should be emphasized that t_A and t_B can be directly estimated from molecular dynamics when simple pore geometries are considered. However, such calculations turn out to be extremely challenging for disordered porous media because the surface/volume decomposition is a complex ill-defined problem. Energy-based criteria such as surface-fluid energy cutoffs or geometrical criteria such as positions to the interface can be used but they rely on arbitrary choices. In contrast, the approach proposed in the present work provides a means to split the complex diffusivity behavior into residence and relocation steps without having to rely on these arbitrary choices.*"

Remarks and questions on Bridging molecular/mesoscopic dynamics in disordered media

Remark 1. *The authors could compare the value of α (reported in Figure 4 b) to the average number of solid atoms encountered in the sphere of first neighbors of methane molecules by calculating the radial distribution function. This would allow them to quantitatively validate their interpretation of the result obtained for α .*

Reply 1. We thank the reviewer for this other interesting point. Indeed, while α should be seen as an effective parameter in our simple modeling of the residence time t_A , we agree with the reviewer that it must be related to the physical number of carbon atoms located around the adsorbed methane molecules. To address this important comment, following the reviewer's suggestion, we have done the following calculations which are included/discussed in the revised version of our manuscript (Page 17): *"To validate this interpretation, we calculated for all interaction strengths $\varepsilon/k_B T$ and porous materials CS_x , the radial distribution function $g(r)$ between host carbon atoms and methane molecules. The number of local carbon neighbors N_c contributing to the free energy barrier involved in the escape time from surface residence was then estimated by integrating $g(r)$ up to the location corresponding to the Lennard-Jones potential minimum $r_{\min} = 2^{1/6}\sigma$, i.e. $N_c = \int_0^{r_{\min}} 4\pi r^2 g(r) \rho dr$. Considering all structures and interactions strengths, we found $\langle N_c \rangle = 3.6 \pm 1$ which is consistent with the value obtained for α in Fig. 4b."*

Remark 2. How do the authors explain that what they call the minimal relocation loop, x_{\min} , is independent of the considered structure ?

Reply 2. We thank the reviewer for raising this very interesting question. In the initial manuscript, taking the minimum loop x_{\min} as independent of the pore structure, it was estimated using Eq. (8). As shown in Fig. 4(d), such a simple estimate captures the behavior of the relocation time as a function of pore size. Following the reviewer's comment, a refined analysis - added to our revised manuscript and detailed in the SI - was undertaken. x_{\min} can be estimated by analyzing the probability density function (PDF) of bridge displacement $\theta(r)$ where r is the end-to-end Euclidean of a Brownian Bridge [see Fig. S10(a) in the revised SI]. The properties of this PDF are discussed in the reference 'Brownian flights over a fractal nest and first-passage statistics on irregular surfaces, Levitz P et al., Phys. Rev. Lett. 96, 180601 (2006)' (which has been added to the manuscript). With this refined analysis, as shown in Fig. S10(b), x_{\min} does depend on the pore diameter d as correctly assumed by the reviewer. Taking into account this evolution, the simulated data $t_B D_s^p$ in Fig. 4(b) as a function of d can be retrieved using a unique value $\beta = 0.7$ [for all values $\epsilon/k_B T$, see Fig. S10(c)]. To address this comment, we have rephrased and extended the following discussion (Page 18): "As shown in Fig. 4(d), by assuming that x_{\min} is independent of the pore structure, Eq. (8) provides a reasonable description of the observed scaling $t_B \sim d$ with a negative intercept in $d = 0$. Yet, as detailed in the Supplementary Information, x_{\min} can be estimated from the probability density function of the bridge displacement $\theta(r)$ where r the is the end-to-end Euclidean of a Brownian bridge \cite{levitz_brownian_2006} [see Fig. S10(a)]. With this refined analysis, as shown in Fig. S10(b), x_{\min} does depend on the pore diameter d . Taking into account this dependence, the simulated data $t_B D_s^p$ in Fig. 4(b) as a function of d can be retrieved using a unique value $\beta = 0.7$ [for all values $\epsilon/k_B T$ as shown in Fig. S10(c)]."

To further address this important point raised by the reviewer, we have added the following discussion together with Figs. S10(a-c) in the Supplementary Information (Page 4, SI): "On the one hand, t_c is associated with a geometrical cut-off length r_c corresponding to the maximal bridge extension. r_c , which is of the order of the pore size d , writes $r_c = \beta d$ with $\beta \sim 1$. On the other hand, t_0 is related to the short-range threshold time for the scaling $t^{1/2}$ associated to the typical distance x_{\min} . Assuming Fickian diffusion upon relocation, we can write $t_0 \sim x_{\min}^2 / 2D_s^p$ and $t_c \sim \beta^2 d^2 / 2D_s^p$. As discussed in the main text, a simple estimation able to capture the overall data behavior in Fig. 4d (obtained for $\epsilon/k_B T = 1$) with a scaling $t_B(d) \sim d$ and a negative intercept $t_B(0) < 0$ leads to $x_{\min} \sim 0.12$ nm and $\beta \sim 0.4$. With this two-parameter fit, x_{\min} is considered independent of the pore structure. Yet, a more advanced analysis can be performed by inspecting the probability density function (PDF) of the bridge displacement $\theta(r)$ with r the end-to-end Euclidean distance of a Brownian bridge \cite{levitz_brownian_2006} [see Fig. S10(a)]. As shown in Fig. S10(a), the bridge displacement follows a power law, i.e. $\theta(r) \sim r^{-2}$, in the intermediate range which is typical of flat surfaces. While an exponential cutoff is observed at large distances, a maximum is observed associated with the end of the algebraic regime at short distances. Using this maximum to define x_{\min} , Fig. S10(b) shows that x_{\min} depends on pore diameter d with a nearly linear increase for d varying from 0.02 to 0.05 nm. As shown in

Fig. S10(c), taking into account this evolution, the generic behavior for $t_B \times D_s^p$ as a function of d can be retrieved with a unique value $\beta = 0.7$ (for all $\epsilon/k_B T$).

Miscellaneous comments

Remark 1. *The stop and go brownian motion described in this study is similar to that of water in cellulose-like materials as reported in a paper by Kulasinski et al. The authors might consider adding this reference to their list.*

Reply 1. We thank the reviewer for mentioning this interesting reference which has been added in the introduction of our revised manuscript - Ref. [36]. We also refer the reviewer to our reply to General Comment 3 by Reviewer 1 which also deals with existing literature on stop-and-go processes.

Remark 2. *This paper deals with gas diffusion in amorphous carbons. It does not seem to me that diffusion in a bulk gas is an activated process. What is the activation energy ΔF_0 of the bulk diffusion coefficient D_{s0} they are referring to ?*

Reply 2. We agree with the reviewer that this point should be clarified. As already suggested in our reply to Remark 5 from the same reviewer (section “Remarks and questions regarding the local diffusion model”), here D_s^0 refers to the bulk diffusivity taken at the same temperature but also at the same density. Therefore, while we agree that the diffusivity of the low density gas phase at the same temperature does not involve any activation energy, the situation is different here as we consider the diffusivity of a dense - liquid-like - phase. To address this comment, we have added the following sentences in the revised manuscript (Page 11): “Here, we refer to the bulk phase taken at the same temperature but also the same density as the confined phase. Therefore, even if the bulk phase is a low density gas (for which diffusion does not involve any activation energy), D_s^0 should be understood as the liquid-like diffusivity of the bulk fluid taken at the same liquid-like density.”

Remark 3. $D_s(r) = D_{s0} \exp(\varphi(r)/kT)$ because $D_s(r) < D_{s0}$ when $\varphi(r)$ is negative.

Reply 3. We thank the reviewer for pointing out this inconsistency in our equation which has been corrected in the revised manuscript. Moreover, we understand from the reviewer’s comment that the sign of the interaction potential should be discussed to avoid any possible misunderstanding. Therefore, to further address this comment, we have slightly modified the following discussion on page 10: “For a confined fluid, the activation energy for diffusion can be assumed to correspond to the bulk activation energy augmented by the fluid/surface potential $\zeta(r)$, $\Delta F = \Delta F^0 - \zeta(r)$ (the sign minus is due to the fact that the interaction potential is attractive and, hence, negative so that molecules are trapped in deeper energy sites with an escape time requiring a larger activation energy)”.

Remark 4. *The authors should use different notations for the porosity, φ , and the effective solid/fluid potential $\varphi(r)$ found at page 9.*

Reply 4. We thank the reviewer for noticing this inappropriate use of the symbol φ . The fluid/solid potential notation was changed to ζ in Page 9 to avoid any possible confusion.

Remark 5. *The label and the caption in figure 4 (a) should state that $t_A = t_{0A} \exp(\alpha\varepsilon/kT)$. Otherwise t_A would increase with temperature and decrease with ε .*

Reply 5. We apologize for the inconvenience. This typo has been corrected in the revised manuscript (both the Figure caption and Figure 4a were corrected).

Reviewer #3

The present paper aims to use the intermittent surface/pore diffusion formalism to map molecular dynamics onto random walk in disordered nanoporous media. I have found particularly interesting the fact that the authors try to link systematically fluid dynamics concepts (e.g. surface residence, in-pore relocation, and their time constants) to underlying simple parameters (e.g. pore curvature and temperature-rescaled surface interaction). The paper is very interesting and definitely well written. Hence I definitively support the publication of this manuscript, after the following (optional) comments will be considered.

Comment 1. *The authors insist on the fact that the decomposition into surface and bulk-like diffusions is sensitive to the exact scaling defined in Eq. (2). Moreover they found that the critical distance for surface diffusion roughly corresponds to the fluid molecular size. All these findings reminded me about another (volumetric) scaling which was proposed some years ago (doi: 10.1038/ncomms4565). I was wondering if the latter could help in quantifying the thickness of the surface region where diffusion is strongly affected by pore interactions.*

Reply 1. We thank the reviewer for this very relevant comment as well as for bringing this interesting reference to our attention. We agree that the model reported in this paper is also efficient at describing the local, i.e. in-pore, diffusivity. To address this important comment, we have extended the following discussion and added the reference by Chiavazzo et al. (Page 11): “While the combination rule above provides a quantitative description of the molecular dynamics data, it remains mostly effective as it relies on arbitrary choices combined with an empirical description of the diffusivity landscape explored by the fluid molecules. First, $\zeta(r)$ should be seen as an effective free energy field that modulates the bulk self-diffusivity by accounting for local intermolecular interactions but also for local density/packing effects. Therefore, even with simple pore geometries, instead of a robust free energy field rigorously derived from intermolecular interactions, $\zeta(r)$ is an effective function which is used to describe the self-diffusivity decay upon increasing the distance to the pore surface. The constant surface diffusivity at the pore surface is used to account for the fact that adsorbed molecules explore homogeneously the surface region $\sim 2\sigma$. Moreover, even if the conclusions above are qualitatively independent of the different assumptions involved, the decomposition into surface and bulk-like diffusions is also sensitive to the exact scaling defined in Eq. (2) and the parameter 2σ used to define the surface layer. In particular, other efficient decomposition rules

have been proposed such as a simple weighted sum of surface and volume diffusivities which was found to accurately describe the dynamics of water in nanoconfinement \cite{chiavazzo_scaling_2014}.”

Comment 2. *The authors claim that the present approach provides a robust formalism to predict diffusion for any fluid in complex nanoporous media using fluid and material parameters available to simple experiments. It would be a significant benefit to report a simple example about practical implications in a real application.*

Reply 2. We agree with the reviewer that practical implications should be better stated. To address this comment, we have rephrased the following discussion in the revised version of the manuscript (Section “Discussion”, Page 19): “Such upscaling strategy could prove to be useful in numerous fields involving fluid adsorption and transport in porous materials: chemistry (e.g. adsorption, catalysis), chemical engineering (e.g. separation, chromatography), geosciences (e.g. pollutant transport), etc. In particular, among important examples relevant to such practical fields, the present approach can help describe molecular diffusion in the following applications: phase separation of gaseous or liquid effluents through porous media, filtration of small micropollutants such as organic/bio molecules, metallic and ionic complexes in water remediation, kinetics of products, reactants and by-products in catalytic processes, etc.”

Comment 3. *I am not sure if the authors used an isotropic molecular model for the methane molecules in the MD simulations. If this is the case, I was wondering what would be the impact of molecule anisotropy on the surface residence and relocation times.*

Reply 3. Indeed, an isotropic molecular model - known as the united atom model - was used to describe the methane molecule. We agree with the reviewer that such information should be specified and that possible extension to more complex fluids should be discussed. To address this comment, we have added the following discussion in the revised manuscript (Page 6): “An isotropic molecular model - known as the united atom model - was used to describe the methane molecule. Such a simplified model was selected as it simply corresponds to a Lennard-Jones potential that is representative of a broad class of atomic and molecular liquids. Despite this simple fluid hypothesis, we believe that our approach can be extended to more complex fluids such as dipolar molecules. In particular, even if complex molecular structures lead to richer surface thermodynamics behavior with strong adsorption in specific sites and/or relocation with large inherent activation energies, the present approach remains relevant as such complexity is embedded - at least in an effective fashion - into the mean relocation and residence times.”

REVIEWERS' COMMENTS

Reviewer #2 (Remarks to the Author):

Dear publisher, dear authors,

I have read the revised version of the article. I am grateful to the authors who took care to answer my questions point by point and in a precise manner.

I am pleased to recommend the publication of this new version in the journal Nature Communications.

Yours sincerely,
Romain Vermorel